# Implementation of a dry deposition module (DEPAC v3.11_ext) in a large eddy simulation code (DALES v4.4)

Leon Geers[1,*], Ruud Janssen[1,*], Gudrun Thorkelsdottir[1,2,3], Jordi Vilà-Guerau de Arellano[2], and Martijn Schaap[1]

[1]TNO, Department of Air quality and Emissions Research, Utrecht, the Netherlands
[2]Meteorology and Air Quality Section, Wageningen University, The Netherlands
[3]Now at: RIVM National Institute for Public Health and the Environment, Center for Environmental Quality, Bilthoven, The Netherlands
[*]These authors contributed equally to this work.

**Correspondence:** Ruud Janssen (ruud.janssen@tno.nl)

**Abstract.** High-resolution data on reactive nitrogen deposition are needed to inform cost-effective policies. Large eddy simulation models coupled to a dry deposition module present a valuable tool for obtaining these high-resolution data. In this paper we describe the implementation of a dry deposition module, that is an extension of DEPAC v3.11 with codeposition (DEPAC v3.11_ext; hereafter simply referred to with DEPAC), into a large eddy simulation code (DALES v4.4), and its first application in a real-world case study. With this coupled model, we are able to represent the turbulent surface-atmosphere exchange of passive and reactive tracers at the hectometer resolution. A land surface module was implemented to solve the surface energy budget and provide detailed information for the calculation of deposition fluxes per land use (LU) class. Both the land surface model and the dry deposition module are extensively described, as well as the inputs that are needed to run them.

To show the advantages of this new modeling approach, we present a case study for the city of Eindhoven in the Netherlands, focusing on the emission, dispersion and deposition of $NO_x$ and $NH_3$. We find that DALES is able to reproduce the main features of the boundary layer development and the diurnal cycle of local meteorology well, with the exception of the evening transition. DALES calculates the dispersion and deposition of $NO_x$ and $NH_3$ in great spatial detail, clearly showing the influence of local LU patterns on small-scale transport, removal efficiencies and mixing characteristics.

## 1 Introduction

Eutrophication and acidification due to atmospheric deposition of reactive nitrogen have widespread impacts on biodiversity (Bobbink et al., 2010; Dise et al., 2011). Across Europe critical nitrogen loads are widely exceeded in protected nature areas, most notably in or close to source regions of ammonia and nitrogen oxides (Jonson et al., 2022). The Netherlands is a densely populated country and has a very productive agricultural sector, a combination leading to the largest emission density of reactive nitrogen compounds in Europe (EMEP/CEIP, 2023). Often, activities emitting large quantities of reactive nitrogen are located in close proximity to nature areas. For example, many farms are located within a kilometer of a nature reserve. In addition,

major roads and highways may cut through or circumvent nature reserves and some of the country's largest industrial facilities border areas declared a nature preservation area. These activities all contribute to total deposition loads on nature areas.

Deposition loads can be seen as a sum of a background deposition caused by a large number of small contributions from distant sources and a local deposition due to nearby sources. Traditionally, these spatial scales have been addressed by different modeling techniques applying chemistry transport models (CTMs) at the regional scale and dispersion models at the local-scale. As the CTMs simulate explicit chemistry on an hour-by-hour basis on a regular grid, they assume instantaneous mixing within their grid cells and thus cannot be used to address near source dispersion and chemistry. In contrast, local scale modeling is often performed with Gaussian plume models driven by statistical meteorological data. These models describe hourly averaged plumes based on wind characteristics, atmospheric stability and downwind surface roughness. Losses due to deposition and chemistry can only be taken into account through very simplified calculation rules and source depletion terms. A number of models combine a plume approach with a trajectory system with a slightly more elaborate accounting of chemical conversion, e.g. OPS (Sauter et al., 2020) and FRAME (Aleksankina et al., 2018; Singles et al., 1998). Both types of models thus have their limitations in representing processes relevant to deposition at the local scale.

Aforementioned local modeling practices prevail due to modest calculation requirements in many (regulatory) applications such as permitting practices. However, a number of applications call for the development of more detailed modeling systems for the local scale. Firstly, for reactive species the processes of turbulent dispersion, chemistry, and deposition take place on similar time and spatial scales and together determine the deposition patterns in a complex landscape. Atmospheric chemistry is often parameterized in these models, which leads to systematic biases in conditions that deviate from the photo-stationary state, especially for fast reacting species such as $NO_2$ (Vilà-Guerau de Arellano et al., 1990; Grylls et al., 2019; Zhong et al., 2017). Besides the photo-stationary equilibrium, the $NO_2$ atmospheric lifetime of 2 to 12 hours may cause a substantial fraction of the locally emitted $NO_x$ to be converted into nitric acid which deposits efficiently. Moreover, the turbulent mixing time scales have been shown to impact the formation of ammonium nitrate (Aan de Brugh et al., 2013; Barbaro et al., 2015), which is supported by reports of ammonium nitrate evaporation impacting ammonia flux measurements (Zhang et al., 1995). Hence, to study the deposition of nitrogen compounds in a complex landscape in detail, a model is required that resolves the turbulent structure of the boundary layer, the chemical interactions and the deposition processes. Secondly, modern observation systems reach temporal and spatial resolutions such that detailed modeling is required to optimize the monitoring design and interpret their results. For example, new satellite missions target resolutions of several hundreds of meters, scales at which plumes will be partially resolved (ESA, 2023). Recent flight campaigns already show the meandering plumes from all kinds of large point sources. Fast response instruments are being used to perform mobile measurements traversing through plumes (Twigg et al., 2022). All these observations do not fit a Gaussian plume representation as they observe an instantaneous realization of the plume at hand. To invert emission strengths from these observation systems, more detailed understanding is needed of the plume behavior and of loss terms between source and receptor. Thirdly, the societal debate on reactive nitrogen deposition and the potential mitigation strategies, as well as the underlying science, calls for the evaluation of the highly parameterized dispersion models that are currently used in many regulatory applications.

Models that resolve the turbulent flow and chemical reactions simultaneously are crucial at spatial scales of 100 m and below. Large Eddy Simulation (LES) models in principle have that capacity, since they resolve the atmospheric transport by the largest scales of atmospheric turbulence. They have been developed since the 1970s (Deardorff, 1970) and have been applied mostly to academic cases. In LES modeling, the turbulent flow field defined by the full Navier-Stokes equations is resolved down to certain minimum length and time scales. Below these scales, a sub-grid scale model (SGS) dictates the behavior of the small scale turbulent eddies and the dissipation of turbulent kinetic energy. In that sense, LES offers many benefits above widely applied Reynolds-averaged Navier-Stokes calculations, which only provide time-averaged properties of fluid flow (Blocken, 2015). Recent developments in computational power and coupling to large scale models (Jansson et al., 2019; Van Stratum et al., 2019; Schalkwijk et al., 2015) have enabled the application of LES on larger spatial and temporal scales than before. Several LES codes are currently being developed towards application in air quality though the development of modules that represent, for instance, the flow around buildings (Tomas et al., 2015), gas-phase chemistry (Vilà-Guerau de Arellano et al., 2011; Kim et al., 2012; Khan et al., 2021) or anthropogenic emissions (Khan et al., 2021). These developments have significantly improved the representation of real-world cases (Maronga et al., 2020; Suter et al., 2022).

Dry deposition of gaseous and aerosol species on the (vegetated) land surface has not been explored much using LES models. Barbaro et al. (2015) have studied gas-aerosol partitioning for ammonium nitrate with DALES, the Dutch Atmospheric Large Eddy Simulation model (Heus et al., 2010). They used a simplified resistance model (e.g. Wesely (1989)) including the aerodynamic resistance, a quasi laminar sub-layer resistance (depending on the molecular diffusivity of the gas) and a bulk surface resistance for a land use consisting of grass only. Clifton and Patton (2021) applied the National Center for Atmospheric Research (NCAR) LES model, coupled to a multi-layer model of the vegetation canopy with a simplified chemical mechanism. They focused on the ozone deposition to canopy and soil, and particularly on the influence of turbulence on the deposition. Subsequently, Clifton et al. (2022) extended the model with a chemical model of 41 reactions and 19 gases for ozone, $NO_x$(= $NO + NO_2$), $HO_x$(=·$OH + HO_2$·), and isoprene chemistry. Although the dry deposition process description has also recently been included in the PALM LES code (Khan et al., 2021), there has not been specific attention to its application.

Our overall goal is to study the process affecting nitrogen deposition at the landscape scale using the DALES system. In this work, we coupled a dry deposition module to the LES code, with the goal of enabling the representation of dry deposition of trace gases over a realistic land use mosaic. Our approach focuses on the application of the well established dry deposition parameterization DEPAC (Deposition of Acidifying Compounds; Van Zanten et al. (2010)) in DALES. To demonstrate the capability of the model to simulate tracer dispersion and dry deposition, we set up a case study over the city of Eindhoven which is the fourth largest city in the Netherlands. In contrast to other large cities in the Netherlands, its inland location ensures that we can use the meteorological forcing as applied here, since there is no coastline or large water bodies in the domain that could induce secondary circulations. The city is surrounded by major highways and an airport, and is embedded in an area with intensive animal husbandry. Hence, it is one of the regions where the reactive nitrogen emissions are greatest and result in exceedance of the critical loads in the nature areas adjacent to the city. In our current analysis, we focus on the dispersion and dry deposition of two passive tracers ($NO_x$ and $NH_3$) and their dependence on atmospheric and LU properties. Chemical conversions are not yet included in this study.

This paper is organized as follows: in Section 2, we describe the DALES model and the DEPAC dry deposition module, as well as the implementation of DEPAC in DALES. Next, we demonstrate the application of the system in a case study for Eindhoven (Section 3). This case study is a first demonstration of dry deposition at high resolution in a complex landscape. Therefore, we will focus on the qualitative aspects and highlight directions for future development. A first evaluation against $NH_3$ concentration and flux observations at Cabauw is included in Section 4. Finally, in Section 5, we summarize our conclusions.

## 2 Model description

### 2.1 DALES

The Dutch Atmospheric Large Eddy Simulation (DALES) is rooted in the LES code of Nieuwstadt and Brost (1986) and Cuijpers and Duynkerke (1993). The model resolves turbulent transport processes based on the filtered Navier-Stokes equations combined with the Boussinesq approximation. DALES uses one-and-a-half-order closure to parameterize subfilter-scale processes. A complete description DALES v3.2 is given by Heus et al. (2010). Since version 3.2, a few additions were made, including improved restart possibilities, new advection schemes, an improved radiation model and optimized subgrid calculations.

DALES has been applied in studies on the transport and chemical conversion of reactive species in the boundary layer. Vinuesa and Vilà-Guerau de Arellano (2003) studied the effect of turbulence on reactive tracer transport, and Vilà-Guerau de Arellano et al. (2005) studied the influence of cumulus clouds on transport of reactive tracers. In a series of case studies over tropical forests, Vilà-Guerau de Arellano et al. (2009) used DALES to study the diurnal cycle of isoprene emissions, Vilà-Guerau de Arellano et al. (2011) performed case studies on the diurnal cycle of concentrations of reactive species, and Ouwersloot et al. (2011) studied the influence of turbulence on the reactivity of chemical species. These studies did not take dry deposition of trace gases into account.

In an idealized DALES experiment, Aan de Brugh et al. (2013) studied the gas-particle partitioning of ammonium nitrate in the convective boundary layer at mid-latitudes. They suggest that turbulent mixing in the CBL causes horizontal and temporal variability in aerosol nitrate mixing ratios, which leads to an apparent downward flux of ammonium nitrate which might be interpreted as nitrate deposition. Barbaro et al. (2015) expanded that case study by including surface-exchange of ammonium nitrate and related gas-phase species in a simplified way. Recently, Schulte et al. (2022) studied the turbulent dispersion of $NH_3$ with DALES, with the goal of assessing the representativity of $NH_3$ observations. In their simulations, they prescribed a constant deposition flux for $NH_3$, based on annual average observations.

### 2.2 Land surface module

The land surface and the atmosphere above it form a tightly coupled system on diurnal (and longer) time scales (Betts, 2004; Van Heerwaarden et al., 2010). The direction and strength of the exchange of heat, moisture, and momentum, is driven by this

coupling. Moreover, land use plays an important role in dry deposition, because the properties of the land surface regulate the uptake of pollutants. Therefore, the relationship between deposition fluxes of reactive compounds, the (vegetated) land surface, and the heat and moisture fluxes is integral to our approach of coupling a deposition module to DALES.

The state of the vegetation, the soil and their exchanges of heat and water with the atmosphere in DALES is described by a Land Surface Model (LSM) that is based on the Tiled ECMWF Scheme for Surface Exchanges over Land with revised hydrology (HTESSEL) land surface scheme (ECMWF, 2021; Balsamo et al., 2009). In HTESSEL, sub-grid surface types are represented by tiles. Each tile represents a LU type for which an energy balance equation is solved. This results in a sensible (H) and latent (LE) heat flux per tile, which is then aggregated to a grid cell. Soil moisture content and soil temperature are calculated in a 4-layer soil model. Precipitation and dew fall is collected in an interception layer. Canopy conductance can be calculated in two different ways in HTESSEL: following the Jarvis Stewart approach or following a plant physiology based approach (A-gs). Here, we followed the former approach. Monin-Obukhov Similarity Theory (MOST) is used to estimate the resistances associated with transport between the lowest model level and the land surface.

We made modifications in the LSM to enable the coupling with the deposition module. Most importantly, we made the number and types of land use classes available flexible. In that way, the land use (LU) types relevant for dry deposition can be selected flexibly, and relevant properties can be assigned. Here, we aligned the LU types with those available in DEPAC (see Table B1).

## 2.3 The DEPAC deposition module

### 2.3.1 Description of the deposition module

To account for dry deposition in local air quality models, one of two approaches is generally followed. The first and simplest approach is the assumption of a constant deposition velocity that is tabulated for each species. This approach has three important downsides. First, there is a significant influence of the wind speed and the turbulence generated near the surface on the transport of species to the surface which is not covered when assuming constant deposition velocity. It is therefore important to incorporate the surface roughness and the surface type in the calculation of dry deposition of both gaseous and aerosol species. Second, variations in plant physiology, temperature, relative humidity and radiation play an important role in determining the dry deposition velocity (see for instance Emberson et al. (2000b)), whereas a constant deposition velocity only contains static influences. Third, the concept of a constant deposition velocity is also inappropriate since it assumes a constant deposition velocity with height near the surface which is not holding for chemically reactive species like $NO_x$ (Vinuesa and Vilà-Guerau de Arellano, 2003). Including these parameters in the parametrization of dry deposition is essential to get insight into diurnal and seasonal cycles, the effect of meteorology on deposition, and to get a realistic representation of atmospheric concentrations and deposition.

The second and most widely used approach, is the resistance model attributed to Wesely (1989), who modelled the deposition velocity $V_d$ (m s$^{-1}$) as the reciprocal sum of three resistances: the aerodynamic resistance $R_a$ (s m$^{-1}$), the quasi-laminar sub-layer resistance $R_b$ (s m$^{-1}$), and the canopy resistance $R_c$ (s m$^{-1}$).

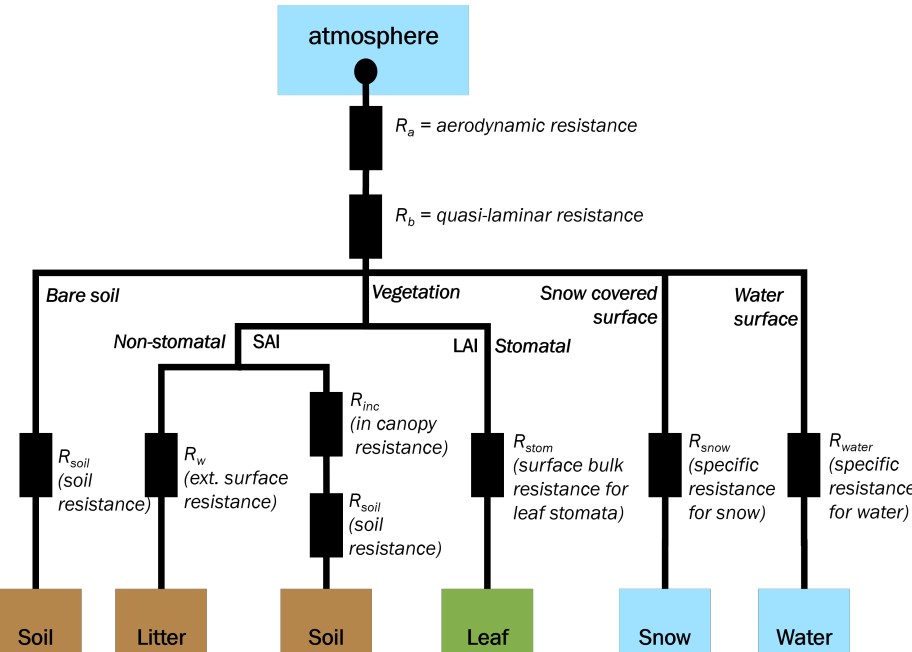

**Figure 1.** Schematic representation of the dry deposition model in DALES.

$$V_d = \frac{1}{R_a + R_b + R_c} \tag{1}$$

The DEPAC deposition module applied in this work is an implementation of the resistance model. A schematic representation of the model is shown in Figure 1, with its deposition pathways to vegetated and non-vegetated surfaces, each with their own specific parameters. It offers a fine grained differentiation of deposition on 9 different land use classes.

The DEPAC module is a well-established module for dry deposition calculations. It is used as a dry deposition module in the air quality models LOTOS-EUROS (Manders et al., 2017, 2022) and OPS (Sauter et al., 2020). A theoretical analysis of
the sensitivity of DEPAC to several of its input parameters is given by Van Zanten et al. (2010). Further, it has been evaluated against observed deposition fluxes over forests (Melman et al., 2025; Wintjen et al., 2022) and dune ecosystems (Jongenelen et al., 2025; Vendel et al., 2023). These analyses have shown that the parameterizations of compensation point and the external resistance ($R_w$) contribute most to uncertainties in calculated deposition (and emission) fluxes of $NH_3$.

### 2.3.2   Calculation of resistances

The calculation of the aerodynamic resistance $R_a$ is based on the meteorological conditions and the surface roughness $z_0$ (m). $R_a$ is common to all species and determined from the stability of the atmosphere and the surface roughness. Because the surface roughness depends on the land use type, in DALES we calculate the $R_a$ per land use type in a grid cell.

$$R_a = \frac{1}{u^* \cdot \Phi_h} \tag{2}$$

where $u^*$ is the friction velocity (m s$^{-1}$) and $\Phi_h$ is a dimensionless parameter that depends on atmospheric stability and is calculated from MOST (Businger et al., 1971).

The quasi-laminar resistance $R_b$ is determined from the friction velocity and the molecular diffusivity of the species in the air.

$$R_b = \frac{2}{\kappa \cdot u^*} \cdot \left(\frac{\mathrm{Sc}}{\mathrm{Pr}}\right)^{(2/3)} \tag{3}$$

in which $\kappa$ is the Von Kármán constant (-), Sc is the dimensionless Schmidt number (-), defined as the ratio between the kinematic viscosity of the air and the mass diffusivity of the species in the the air, and Pr is the dimensionless Prandtl number (-), defined as the ratio between the kinematic viscosity of the air and the heat diffusivity of the air (definition according to Wesely and Hicks (1977)). Although there is a species dependence (in Sc/Pr) in $R_b$, it usually plays a minor role, because the quasi-laminar layer is usually very thin (of the order $\nu/u^*$, where $\nu$ is the kinematic viscosity of air) and $R_b$ is therefore small. In the calculation of $R_b$, the molecular diffusivity and viscosity are used.

The canopy resistance $R_c$ is the resultant of three parallel resistances: the stomatal resistance, the soil resistance and the external surface resistance. The canopy resistance $R_c$ is calculated from stomatal and non-stomatal conductances and resistances based on the model of Erisman et al. (1994).

$$R_c = \left(\frac{1}{R_{stom}} + \frac{1}{R_{soil,eff}} + \frac{1}{R_w}\right)^{-1} \tag{4}$$

Non-stomatal transport is divided into transport to external surfaces (cuticles and other surfaces, represented by resistance $R_w$ in s m$^{-1}$) and transport through the canopy to the soil ($R_{soil,eff} = R_{inc} + R_{soil}$ in s m$^{-1}$). For the latter, a distinction is made between snow-covered and open soil. The stomatal resistance $R_{stom}$ (s m$^{-1}$) is implemented according to Emberson et al. (2001). The stomatal conductance is controlled by the type of vegetation, its phenological state, temperature, relative humidity, insolation, and the potential for water of the soil.

In general, there are two main variants of this resistance approach. The first assumes a concentration of species on the deposition surface of zero; the deposition flux $F_{dep}$ (μg m$^{-2}$ s$^{-1}$) is then modelled as the product of the deposition velocity and the concentration in the atmosphere $\chi_{atm}$ (μg m$^{-3}$) (e.g. Simpson et al., 2012). Although it performs well for compounds that are completely absorbed through the surface, this approach induces possible overestimation of the deposition fluxes and does not anticipate re-emission of species from the surface into the atmosphere. Especially for NH$_3$ during growing seasons, this can result in unrealistic deposition fluxes (Wichink Kruit et al., 2007).

The second variant of the resistance approach considers a non-zero concentration on the deposition surface, called a compensation point $\chi_{comp}$ (μg m$^{-3}$). This is the concentration that is in equilibrium with the reservoir of the species, in this case

ammonium, under the surface. When ammonia concentrations in the atmosphere are higher than the compensation point, deposition occurs. Emission occurs when the atmospheric ammonia concentration is below the compensation point. Hence, the exchange is bi-directional. The resulting flux is calculated with:

$$F_{dep} = -V_d \cdot (\chi_{atm} - \chi_{comp}) \tag{5}$$

The compensation point is defined for the stomatal and external surface deposition routes, and its parametrization is given in Section 2.3.3.

At this moment, the deposition model uses a dedicated land surface model that is different from HTESSEL. While HTESSEL is used for the calculation of energy and moisture fluxes, the land surface model in the deposition model is dedicated to the deposition flux calculations. Future development plans include the integration of these two modules, but at this moment, they are used in parallel. In dry conditions, this is not a problem, but there is a caveat to this approach regarding wet surfaces. HTESSEL defines a dedicated land surface class for wet surfaces, regardless the underlying land surface. The land surface model for deposition uses a flag for each land surface type that signals whether it is dry or not. Due to these different implementations, cases with wet soil and/or canopy cannot be simulated properly, as the current implementation of the deposition model ignores the fact that the soil/canopy is wet (it assumes dry conditions). In our case study, we circumvented this problem, since the KNMI weather data at the location of Eindhoven airport showed that the date of our simulations was preceded by a prolonged period of dry conditions (11 days without any precipitation).

### 2.3.3 Canopy exchange

The first term on the right in Equation 4, the deposition resistance for stomatal exchange, is defined in Equation 6 below. Per LU class, a maximum canopy conductivity $G_s^{max}$ (m s$^{-1}$) is defined, which signifies the conductivity in the case of fully opened leaf stomata. The stomata may not to be fully open due to a number of causes; for each of these, a dimensionless parameter between zero and one is multiplied with $G_s^{max}$ to scale the transport. $f_{phen}$ is the correction factor for plant phenology, $f_{swp}$ corrects for the soil water potential, $f_{vpd}$ for the vapour pressure deficit, $f_T$ is a temperature correction, and $f_{PAR}$ is the correction for photo-active radiation (Emberson et al., 2000b, a).

$$R_{stom} = (G_s^{max} \cdot f_{phen} \cdot f_{swp} \cdot f_{vpd} \cdot f_T \cdot f_{PAR})^{-1} \tag{6}$$

where the correction factor for phenology $f_{phen}$ and the correction factor for soil water potential $f_{swp}$ are both assumed equal to 1.0. This assumption is justified as follows: for the current LU classes, the impact of phenology is minimal (Van Zanten et al., 2010). Instead, the seasonal variation in vegetation phenology is accounted for by using a leaf area index (LAI in $m_{leaf}^2\ m_{ground\ surface}^{-2}$) that changes according to the vegetation's growing season.

$$G_s^{max} = g_s^{max} \cdot LAI \tag{7}$$

where $g_s^{max}$ is the maximum leaf conductance ($ms^{-1}$). Similarly, the influence of soil water potential in North-Western Europe is expected to be limited (Van Zanten et al., 2010).

Specifically for $NH_3$, the stomatal compensation point $\chi_s$ (µg m$^{-3}$) is calculated from an equilibrium correlation (Wichink Kruit et al., 2007). $\Gamma_s$ is the dimensionless ratio between the apoplastic molar $NH_4^+$ and $H^+$ concentrations given by (Wichink Kruit et al., 2010):

$$\chi_s = \frac{2.75 \cdot 10^{15}}{T_s + 273.15} \exp\left(\frac{-1.04 \cdot 10^4}{T_s + 273.15}\right) \cdot \Gamma_s(T_s) \tag{8}$$

$$\Gamma_s(T_s) = 362 \cdot \chi_{a,4m,long\ term} \cdot 4.7 \cdot \exp(-0.071 T_s) \tag{9}$$

in which $\chi_{a,4m,long\ term}$ is the 'long term' mean concentration of $NH_3$ at 4m above the ground (µg m$^{-3}$) and $T_s$ is the leaf surface temperature (ºC). In our application, we apply a monthly mean concentration for $\chi_{a,4m,long\ term}$. This approach was chosen, because it is unknown how much of the depositing species is accumulating in the vegetation; there is no mass balance of the vegetation. Instead, the compensation point is estimated by assuming the accumulated amount of a species in the vegetation is in equilibrium with its long term mean atmospheric concentration.

External leaf surface exchange is covered by the resistance $R_w$, as defined in Eq. 10, with $RH$ being the relative humidity (%), and $\alpha = 2$ s m$^{-1}$ and $\beta = 12$ are empirical model parameters (Sutton and Fowler, 1993). The deposition model uses an additional correction factor dependent on the surface area index (SAI in m$^2_{\text{deposition surface}}$ m$^{-2}_{\text{ground surface}}$) to account for differences between SAI of the local vegetation type and the SAI at the measuring site where the compensation point $\chi_w$ was determined (Haarweg, Wageningen, The Netherlands). $T_w$ is the surface temperature (ºC) and $\Gamma_w$ is the dimensionless molar ratio between the $NH_4^+$ and $H^+$ concentrations in the external leaf surface water (Van Zanten et al., 2010; Wichink Kruit et al., 2010).

$$R_w = \frac{SAI_{Haarweg}}{SAI} \cdot \alpha \exp\left(\frac{100 - RH}{\beta}\right) \tag{10}$$

For freezing conditions, a fixed value of $R_w = 200$ s m$^{-1}$ is assumed.

A compensation point can be calculated for the external surfaces analogue to Eq. 8 by substituting $\Gamma_w$ for $\Gamma_s$. Specifically for $NH_3$ over grassland, $\Gamma_w$ is defined according to the empirical relation in Equation 11, in which $\chi_{a,4m}$ is the compensation point at 4 m (Wichink Kruit et al., 2010). For external surfaces, it is assumed that there is always a thermodynamic equilibrium between the surface and its surroundings. Therefore, $\chi_{a,4m}$ has the same value as the atmospheric concentration.

$$\Gamma_w(T_w) = 1.84 \cdot 10^3 \cdot \chi_{a,4m} \cdot \exp(-0.11 T_w) - 850 \tag{11}$$

### 2.3.4 Soil exchange

Depositing species first have to travel through the canopy before they reach the soil. The effective soil resistance is therefore calculated as the sum of the in-canopy resistance $R_{inc}$ and the soil resistance $R_{soil}$:

$R_{soil,eff} = R_{inc} + R_{soil}$ (12)

The in-canopy resistance is a function of LU type, the height of the vegetation $h$ (m), the SAI and the friction velocity $u^*$ (Van Pul et al., 2008):

$$R_{inc} = \begin{cases} \frac{b \cdot h \cdot SAI}{u^*} & u^* > 0,\ \text{arable land, permanent crops, forest} \\ 1000\ \text{s m}^{-1} & u^* = 0,\ \text{arable land, permanent crops, forest} \\ 0\ \text{s m}^{-1} & \text{water, urban, desert} \\ \infty\ \text{s m}^{-1} & \text{grass, other} \end{cases}$$ (13)

in which $b$ is an empirical constant ($14\ \text{m}^{-1}$).

The SAI depends on the LAI and the LU type. Some LU classes have no SAI ($SAI = 0$ for water, barren land, urban), some have a constant value ($SAI = b_{SAI}$ for arable land, grassland and semi-natural land) and for others it depends on the LAI ($SAI = a_{SAI} \cdot LAI + b_{SAI}$ for permanent crops and forests). In case one of the resistances is irrelevant (e.g., in case of stomatal resistance for aqueous surfaces), the resistance value is set to -9999, which is a special value to indicate that the resistance is not to be considered in the calculation of the deposition velocity.

The parameters of the in-canopy resistance are defined for 9 different LU classes (see Table A2). The LAI is a function of the time of year and the growing season per LU class (see Van Zanten et al., 2010).

The soil resistance is species dependent and, to a lesser extent, LU dependent (except for frozen, wet, and snow covered soil). For $NH_3$, $NO_2$, and $NO$, the values are presented in Table A3. These values are based on Erisman et al. (1994), but adapted to the values used in the DEPAC v3.11 implementation in LOTOS-EUROS v2.2 (Manders et al., 2022). Finally, the

compensation point of the soil can be calculated similar to the stomata (Eq. 8) with $\Gamma_{soil}$ substituted for $\Gamma_s$, the value of which is also given in Table A2.

### 2.3.5   Implementation in DALES

In the main driver of DALES, for each time step (i.e. every few seconds), the dry deposition routine is called in a section of surface routines, after the radiation terms are calculated and before advection and diffusion terms are calculated. First, the

necessary parameters like LAI and SAI are calculated based on values from a parameter table. Then the deposition budget is calculated for each depositing tracer by a call to the DryDepos_Gas routine in the dry deposition module. Finally, in a loop over all depositing tracers the deposition budget for each tracer is added to the tracer tendency array in all surface cells.

### 2.4   Initial and boundary conditions

The bottom boundary condition for DALES is formed by the soil and LU properties. Representation of LU properties is

especially important for dry deposition calculations. We use the TOP10NL LU map at 10 x 10 $\text{m}^2$ (PDOK, 2023) that is

translated to the land surface types originally defined in DEPAC according to Table B1. Soil type follows the BOFEK soil map Heinen et al. (2022) and soil hydraulic properties (Van Genuchten, 1980).

For meteorological variables, we apply doubly-periodic boundary conditions, which means that forcings are horizontally homogeneous but vary in time and height (Van Stratum et al., 2023). In this approach, the initial conditions (atmosphere
and soil) and a number of time and height varying large-scale processes acting as forcings on the LES domain, are derived from routine meteorological variables. These forcings, both as state variables like temperature, humidity, or wind and as tendencies of a state variable, are then added to the prognostic LES equations. The surface energy balance calculation is very sensitive to soil moisture and therefore needs to be initialized at realistic value. The open-source Python package "Large-eddy simulation and Single-column model - Large-Scale Dynamics" or (LS)$^2$D (Van Stratum et al., 2023) was used to create
initial and boundary layer profiles as input for DALES (sources available from https://github.com/LS2D/LS2D) from ERA5 reanalysis data (Hersbach et al., 2020).

## 2.5 Anthropogenic emissions

For realistic simulation of passive and reactive tracers at resolutions of tens of meters, detailed information is needed about their emissions at this scale. The emissions of $NO_x$ and $NH_3$ we use are based on the official gridded emissions for the Netherlands
(the Dutch Emission Registration, ER) for 2018 (RIVM, 2018). The official data contains emissions from more than 700 source types, which are aggregated into 15 sectors. Each source type gets assigned a spatial fraction map that is used to downscale the emissions. This fraction map describes the fraction of the emissions in each 1 x 1 km$^2$ grid cell of the inventory that is assigned to the 50 x 50 m$^2$ grid cells of DALES. Hourly emission fluxes are calculated from annual emission totals using fixed monthly, daily and hourly time profiles. This spatial and temporal interpolation, however, creates a large uncertainty in the
emission dataset, which will cause uncertainty in the timing and magnitude of emission peaks on a given day and therefore in the calculation of concentrations.

For the industry, households, and commercial activities the Basisregistratie Adressen en Gebouwen (BAG) is used (Kadaster, 2024). The BAG contains information on locations of buildings, number of addresses in each building, their size, and their function, which we use to link each building to one of these three categories. The number of addresses is used to calculate the
fractions. A national database of the road network (Nationaal Wegenbestand, 2021 (NWB, 2024) was used to make fraction maps for road transport on different road types (highways, main roads, and residential roads), based on the total road length within each grid cell. For agricultural activities, fractions maps were made for livestock, arable land, and general agricultural activities, based on the Basisregistratie Gewaspercelen (BRP) dataset with information on locations of agricultural plots and crop types (NGR, 2024). For rail roads a map of rail road tracks is used (Prorail, 2020) and for inland shipping we use inland
waterways from the same database as the road network (NWB, 2024). Finally, we use a population density map at 100 m resolution for several other source types (CBS, 2023). This resulted in a total of 12 fraction maps.

For the major point sources (mainly industrial), exact locations and emission heights are provided. For non-point sources the emission height is at ground level. For the DALES setup in this work, ground level means the height of the lowest vertical level which is between 0 and 20 m. Plume rise has not yet been implemented in DALES. The effect of this is limited in this study

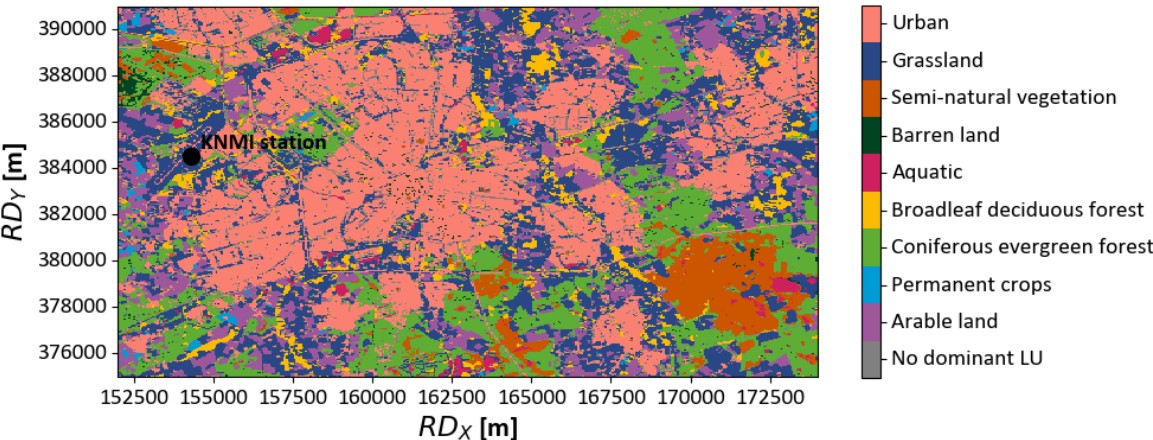

**Figure 2.** Map of the Eindhoven domain showing the land use and the locations of the KNMI measurement station (black dot). Map is plotted using Rijksdriehoeks (RD) coordinates (EPSG 28992).

since there are only a few stacks in the current domain. In the current approach, emissions on city roads appear much less intense than on highways (by a factor of 50, approximately). This is caused by the way emissions are estimated and downscaled in our current emissions preprocessor. It is still under development; work is being done on the development of a more dynamic emission model, which does not imply fixed time profiles, but takes seasonal, meteorological, and environmental factors into consideration. For example, the spatio-temporal variability of ammonia emissions is strongly dependent on agricultural

practices, livestock distribution, crop distribution and meteorology (Ge et al., 2023).

## 3   Eindhoven case study

To demonstrate the ability of DALES coupled to a dry deposition scheme to represent nitrogen transport and deposition at high spatial resolution, we performed simulations for a full diurnal cycle over a domain around the city of Eindhoven (Figure 2). We chose a day with clear-sky convective conditions (16 June 2022). The month of June in 2022 was warm with a lot of sunshine,

but also a lot of rainfall in the Netherlands. The day of our case study fell in a sunny and dry weather period that started the $9^{th}$ of June. The wind direction during this period was mostly from the Northeast, bringing warm and dry air masses. On the day after our case study (June $17^{th}$), temperatures increased to tropical values (above 30ºC). The size of the model domain is 22 x 16 $\text{km}^2$, and it is divided into 440 x 320 grid cells of 50 x 50 $\text{m}^2$ each. In the vertical direction, the domain extends up to 8.5 km and it is divided into 176 levels. Near the surface, the level thickness $\Delta z$ is 20 m, and it gradually increases to 95 m at

the top of the domain according to the equation $\Delta z = \Delta z_0 \cdot (1 + 0.009)^{iz}$ , where $\Delta z_0$ is the depth of the bottom layer and $iz$ the index of the vertical grid.

Hourly observations of standard meteorological variables were available from the KNMI weather station at Eindhoven airport (KNMI, 2024b), station no. 370). The weather data at the location of Eindhoven airport showed that the date of our

simulations was preceded by a prolonged period of dry conditions (11 days without any precipitation). Therefore, it was justi-
fied to use the equations for dry land in the model. Attenuated backscatter profiles from the CHM15k ceilometer at Maastricht
Aachen airport were retrieved from KNMI (KNMI, 2024a). They provide daily aggregated files at five sites in the Netherlands,
Maastricht Aachen airport being the closest to Eindhoven.

The profiles of $NO_x$ and $NH_3$ concentrations were initialized to typical values in a shallow boundary layer, since the
calculations start at night. $NO_x$ was set to 1.0 ppb, from the surface up to 200 m, and 0.5 ppb up to 1250 m high, to simulate
the residual layer of the previous day. For $NH_3$, there is only the 200 m boundary layer with a concentration of 2.7 ppb.
The concentration values are typical background concentrations measured in and around Eindhoven. Finally, it is important
to note that the present calculations do not include atmospheric chemistry yet, so only emission, transport, and deposition are
calculated.

### 3.1 Diurnal cycle, vertical mixing

Calculated sensible heat flux (H) and latent heat flux (LE) data from the DALES simulation were compared to the original
ERA5 data that was used as forcing (see Figure 3). The sensible heat flux becomes positive around 7:00 local time and peaks
around 14:00 LT. Overall, the simulations match ERA5 data well, except for the hours around midday, when the sensible heat
flux is larger by about 100 $\mathrm{Wm^{-2}}$ in DALES, and the latent heat flux is smaller by about 50 $\mathrm{Wm^{-2}}$. A possible explanation
is the difference in local LU properties in DALES compared to the ERA5 dataset. DALES in the current setup includes 9 LU
types (Section 2.4), whereas the ECMWF IFS only includes 5. Moreover, the resolution of the TOP10NL map in DALES is
much higher than the LU map from ECMWF IFS (10 m versus 1 km), so the local landscape is represented in more detail.
This leads to differences in the radiative properties of the surface, which will affect the amount of soil moisture and the surface
energy balance.

The virtual potential temperature ($\theta_v$), specific humidity ($q_t$), and wind speed ($U$) and direction (WDIR) are presented
in Figure 4, together with reanalysis data from the ERA5 dataset. ERA5 data are available on a 0.25º x 0.25º resolution,
corresponding to a 28 x 17 $\mathrm{km^2}$ area around Eindhoven, compared to the present domain averaged DALES results on an area
of 22 x 16 $\mathrm{km^2}$. Downscaling of the meteorological quantities from ERA5 is handled well by DALES, since all quantities
follow the ERA5 data closely, though the wind speed is slightly underestimated.

Local observations from a weather station at Eindhoven airport are also plotted in the figure. Since the wind speed is rounded
to the nearest integer value and the wind direction is rounded to the nearest full degree, the precisions of the measurements
are at least 0.29 $\mathrm{m\,s^{-1}}$ and 3°, respectively. The $\theta_v$ is represented well by DALES between 06:00 and 18:00 LT, but strongly
underestimated before and after sunset. The $q_t$ matches within 0.0015 $\mathrm{kg\,kg^{-1}}$, but there is a 2-3 hr shift of a trough in the
morning. Wind speed shows stronger deviations: up to 2 $\mathrm{m\,s^{-1}}$ speed differences. The wind direction in DALES and ERA5
show an northeast to east direction during most of the day while the observations are between north and northeast.

Vertical profiles of virtual potential temperature ($\theta_v$) show the development of a well-mixed convective boundary layer in
the morning and the formation of a stably stratified nocturnal boundary layer at night (Figure 5a). The wind speed in the mixed

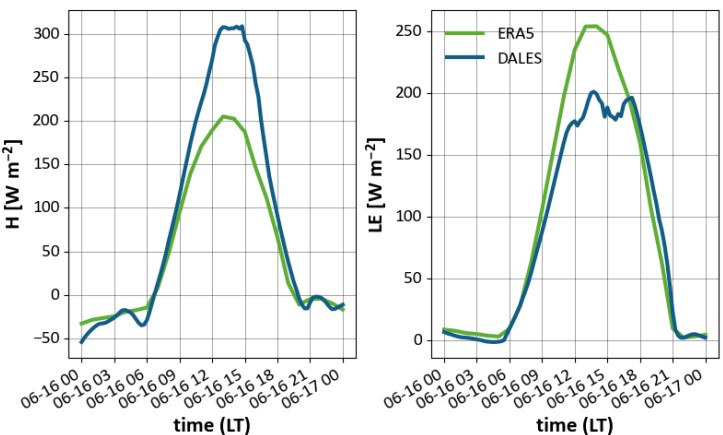

**Figure 3.** Sensible and latent heat fluxes averaged over domain from ERA5 (green) and DALES (blue).

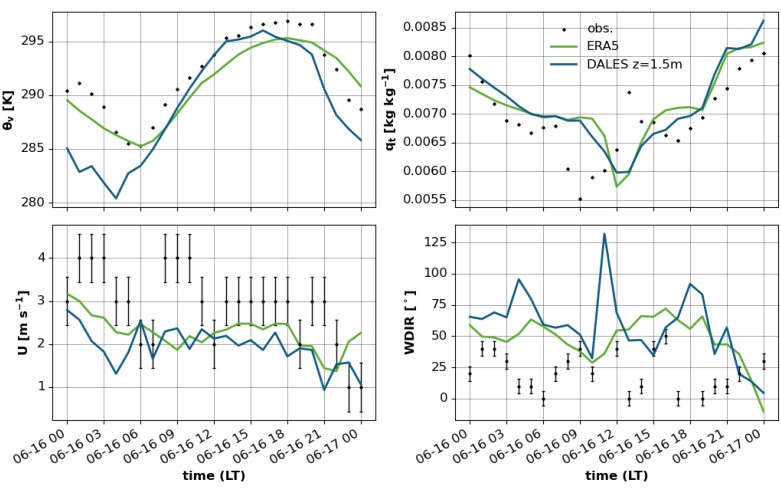

**Figure 4.** Time series of virtual potential temperature, specific humidity, wind speed and wind direction from DALES (blue), ERA5 (green) and observations (dots) on 16 June 2022. DALES data is extrapolated to measurement height (1.5 m).

layer ranges from $4 \, \text{m s}^{-1}$ during the daytime to $7 \, \text{ms}^{-1}$ in the early morning of June $17^{th}$ (Figure 5c). The $\theta_v$-profile indicates a boundary layer evolution from $250 \, \text{m}$ at 06:00 LT to about $1250 \, \text{m}$ at 12:00 LT and below $200 \, \text{m}$ during the night (00:00 LT).

Comparison with ceilometer data at Maastricht Aachen airport is shown in Figure 5d. The colors depict the attenuated back-scatter signal of the ceilometer and the black line is the boundary layer height calculated by DALES. Although the measurement station is approximately $100 \, \text{km}$ from Eindhoven, the trend should be comparable. The development of the boundary layer calculated from DALES fits the ceilometer data well up to about 21:00 LT, after which the modeled boundary layer height shows a sharp decrease, whereas the ceilometer data show a more gradual decrease, taking about 1.5 hours to decrease to its nocturnal value. This is likely related to the evening transition, during which the strength of the convective turbulence decreases, as buoyancy disappears as its driving force, and a shallow nocturnal boundary layer is formed (Darbieu et al., 2015). During this transition, turbulent length scales decrease to sizes that cannot be resolved at the $50 \, \text{m}$ resolution of our simulations. The much smaller eddies that dominate the evening atmosphere will be of comparable or even smaller scale than the cell size, which means that most of the turbulence is covered by the sub-grid scale model. Simulations at $25 \, \text{m}$ resolution (which is the lower bound currently set by the emission downscaling) did not lead to improvements in the representation of the evening transition. A horizontal resolution of $\approx 10 \, \text{m}$ would be more appropriate to simulate the transition to a stable boundary layer (Beare et al., 2006). In addition to using a finer resolution, applying a different sub grid model may help to improve the simulation of the transition to a stable boundary layer. For this study, we applied the SFS-TKE $e$ scheme (Deardorff, 1980), which is the default in DALES. In future studies, the use of a sub-grid model that for the simulation of stable nocturnal boundary layers with coarse resolution (Dai et al., 2021) will be explored. At night, boundary layer heights of 0 m are diagnosed due to the use of the maximum $\Delta\theta_v / \Delta z$ as criterion. This approach is valid under convective conditions only, when the boundary layer is capped by a clear temperature inversion. Figure C1 shows boundary layer heights from the same simulation diagnosed with different approaches.

In conclusion, the evaluation of the meteorological variables shows that the DALES configuration applied in this study is able to downscale the ERA-5 meteorological situation successfully. As expected, the different LU classification induces slight differences in the latent and sensible heat fluxes. The main caveat is that the collapse of the boundary layer is quite sudden, which will negatively impact the modelled pollutant concentrations as the mixing after the evening rush hour is clearly underestimated.

## 3.2 NO$_x$ and NH$_3$ emissions, dispersion and deposition

Maps of the downscaled NO$_x$ and NH$_3$ emissions at a resolution of 50 x 50 $\text{m}^2$ and their time profiles are shown in Figure 6. NO$_x$ emissions are assumed to consist of 97% NO and 3% NO$_2$. The maps show the emissions at 06:00 LT, before the morning rush hour. The NO$_x$ emissions range over 3 orders of magnitude with very low values in the rural areas, increasing in the urban areas moving closer to the city center, to their maximum values on the highways. The time profile shows two clear peaks during the morning and evening rush hours, when the emissions from traffic dominate, intermediate values during the day, and low values at night. As traffic is a relevant source of ammonia in large urban areas (Wen et al., 2023), we can identify the highways as strong emitters of NH$_3$, in a way similar to NO$_x$. Some strong agricultural NH$_3$ sources can also be found in

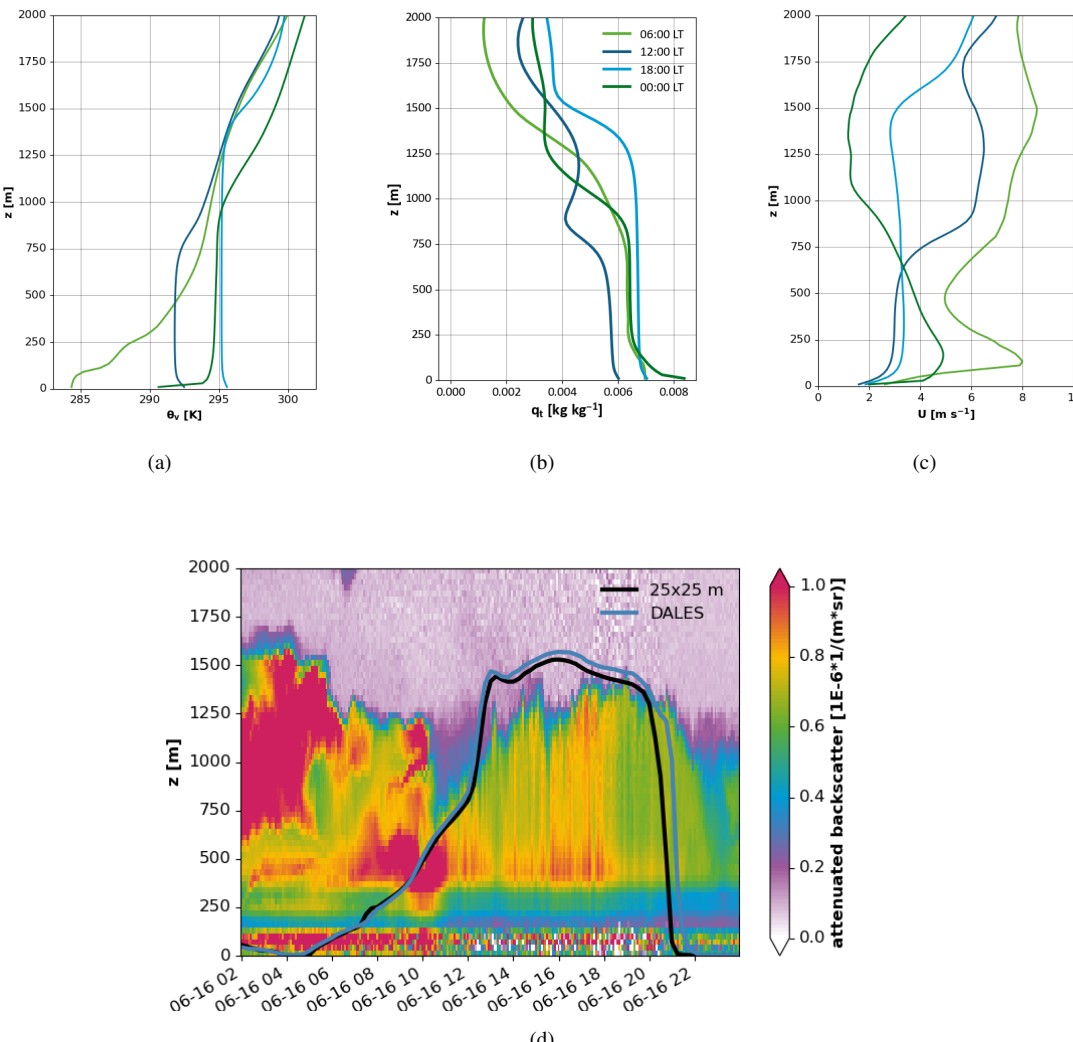

**Figure 5.** Vertical profiles of domain averaged virtual potential temperature $\theta_v$ (a), total specific humidity $q_t$ (b) wind speed $U$ (c) and boundary layer height and ceilometer backscatter profile on 16 June 2022 (d). The latter includes the simulated boundary layer height for the default simulation (DALES; 50 x 50 m$^2$ horizontal resolution) and an additional simulation at 25 x 25 m$^2$ resolution.

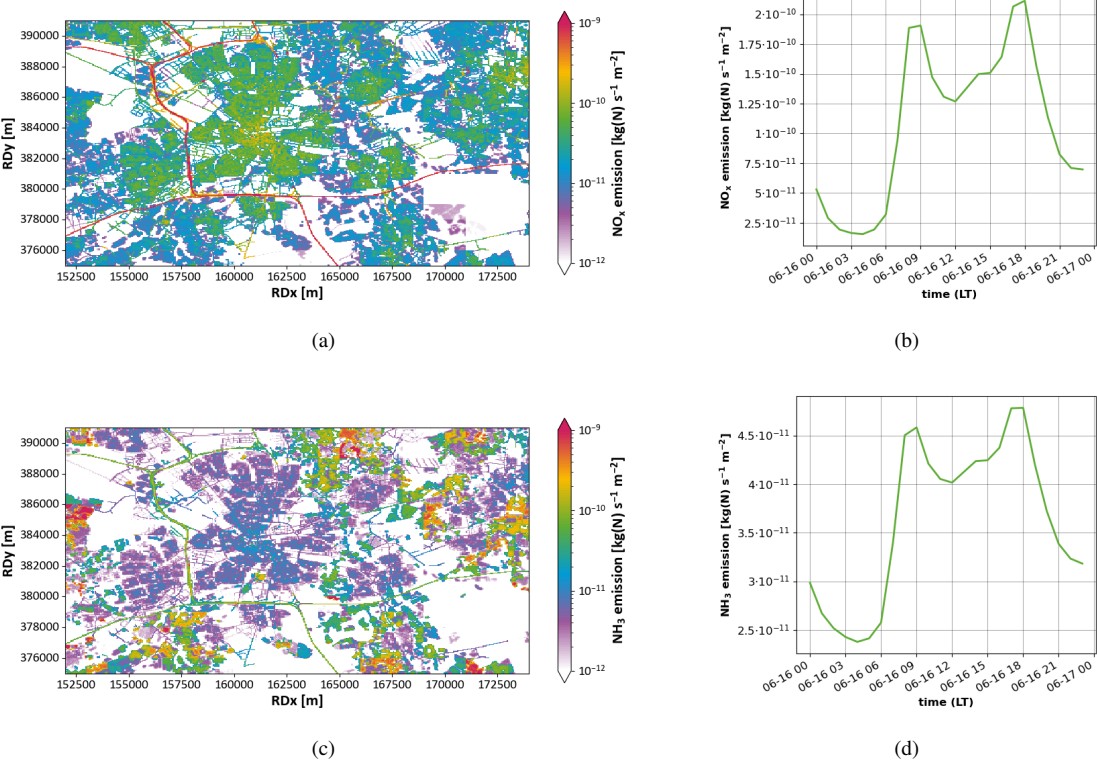

**Figure 6.** Spatial distribution of $NO_x$ (a) and $NH_3$ (c) emissions (at 06:00 LT) and the domain averaged emission time profiles of $NO_x$ (b) and $NH_3$ (d). Maps are plotted using Rijksdriehoeks (RD) coordinates (EPSG 28992).

the rural areas on the western, northeastern, and southeastern edges of the domain. The $NH_3$ emissions show a similar temporal profile as the $NO_x$ emissions, but with values that are about an order of magnitude lower. The time profiles mainly reflect the diurnal cycle in traffic emissions, with rush hour peaks in morning and end of afternoon (Manders et al., 2017). Diurnal cycles for sectors like industry and agriculture have a more flat profile over the day .

The resulting vertical profile of the mean $NO_x$ concentration over the domain (Fig. 7a) shows a trend similar to the boundary layer development (Fig. 5d, Fig. C1). This demonstrates the mixing of the emitted $NO_x$ over the whole boundary layer. The concentration shows a peak near the surface in the morning (09:00 LT), caused by the rush hour emissions (Fig. 6b) and the low boundary layer into which these emissions are mixed (Fig. 5d). A second and much higher peak in the $NO_x$ concentration is simulated in the evening (22:00 LT). This peak coincides with a minimum in the diagnosed boundary layer height, and is

caused by a lack of mixing simulated by DALES in the evening. Improving this will be a priority for future applications of DALES in air quality and deposition studies. Near the top of the convective boundary, $NO_x$ mixing ratios are diluted due to entrainment of $NO_x$ poor air from the free troposphere. The $NO_x$ turbulent flux profile shows a positive flux throughout the boundary layer at all times during the simulation. This means that emission and upward mixing dominate over dry deposition

and entrainment. This is expected for a tracer with strong sources over the whole domain. In addition, the dry deposition flux of $NO_x$ is significantly lower than the dry deposition flux of $NH_3$. The highest fluxes are found at 12:00 and 18:00 LT when both emissions and vertical mixing are strong (Figure 7b).

For $NH_3$, we find a negative concentration gradient from the surface to the boundary layer background concentration of 2.7 ppb at 12:00, 18:00 and 00:00 LT, but a positive gradient at 06:00 LT (Figure 7c). This positive gradient is caused by dry deposition. In the early morning, the uptake of $NH_3$ on wet surfaces forms a strong sink, which leads to a net negative $NH_3$ flux towards the surface (Figure 7d). Fluxes of both $NO_x$ and $NH_3$ show a strong divergence, countering the concept of a constant deposition velocity.

The temporal development of the domain averaged concentrations in the lowest model layer (between 0 and 20 m) (Figure 8) shows similar trends for the two species. Near-surface concentrations are relatively constant during daytime, with a significant increase in the evening. This increase can be explained by the fast collapse of the boundary layer (Figure 5d) and by the fact that the emissions from evening traffic are still high, while atmospheric mixing has already subsided.

The diurnal cycle of $NO_x$ shows a strong relationship between the concentration and the flux (Fig. 8a), indicating that the deposition flux is mainly driven by the concentration gradient between the atmosphere and the surface. It also shows that during most of the day, the canopy resistance ($R_c$) is the most important factor that drives the deposition velocity ($V_d$; Fig. 8c). For $NO_x$, the $R_c$ mostly follows the stomatal resistance, as shown by the decreasing $R_c$ during daytime. Only during nighttime, the aerodynamic resistance ($R_a$) plays a significant role in determining the deposition velocity. For $NH_3$, a different picture emerges (Fig. 8b). During nighttime (between 00:00 and 06:00 LT), the $NH_3$ concentration and deposition are coupled, but during the morning (between 06:00 and 10:00 LT), the concentration increases while the deposition flux decreases. This is driven by an increasing $R_c$ (Fig. 8d). For $NH_3$, the uptake on wet external surfaces is the most important contribution to the canopy uptake, and this pathway decreases when dew on the leaves evaporates. During daytime (10:00 to 20:00 LT), concentration and flux remain relatively constant. In the evening (between 20:00 and 00:00 LT), the deposition flux increases due the increased concentration, while the $R_c$ decreases again due to dew formation. The $V_d$ is dampened by the rising $R_a$. For both species, $R_b$ only plays a minor role in determining the deposition velocities during the whole day.

Figure 9a shows the concentration distribution in the domain at 08:00 LT in the lowest layer of the model. The highways clearly show up as the main sources of $NO_x$, producing a local increment of more than 15 $\mu g\,m^{-3}$ which is dispersed downwind. In addition, a few other strong sources (mainly industrial) are visible at different locations in the domain. The plumes of $NO_x$ generated by highway traffic can still be discerned visibly from the background at 2 - 3 km from the source. The emitted nitrogen oxides are carried downwind, undergoing vertical mixing which causes the near-ground concentrations to reduce away from the source. The atmospheric mixing depends on the LU adjacent to the highways, as the concentrations trail off at a lower rate over surfaces with a low roughness (grass) compared to high roughness (forest). The concept of blending distance can be used to quantify the distance over which emission plumes travel before they are well mixed with the background concentration distribution (Schulte et al., 2022).

The lowest concentrations are found at the East and North border of the domain, and it is the boundary conditions imposed on the model (see Section 3) that explain this. At this moment, DALES does not yet support open boundary conditions. Instead,

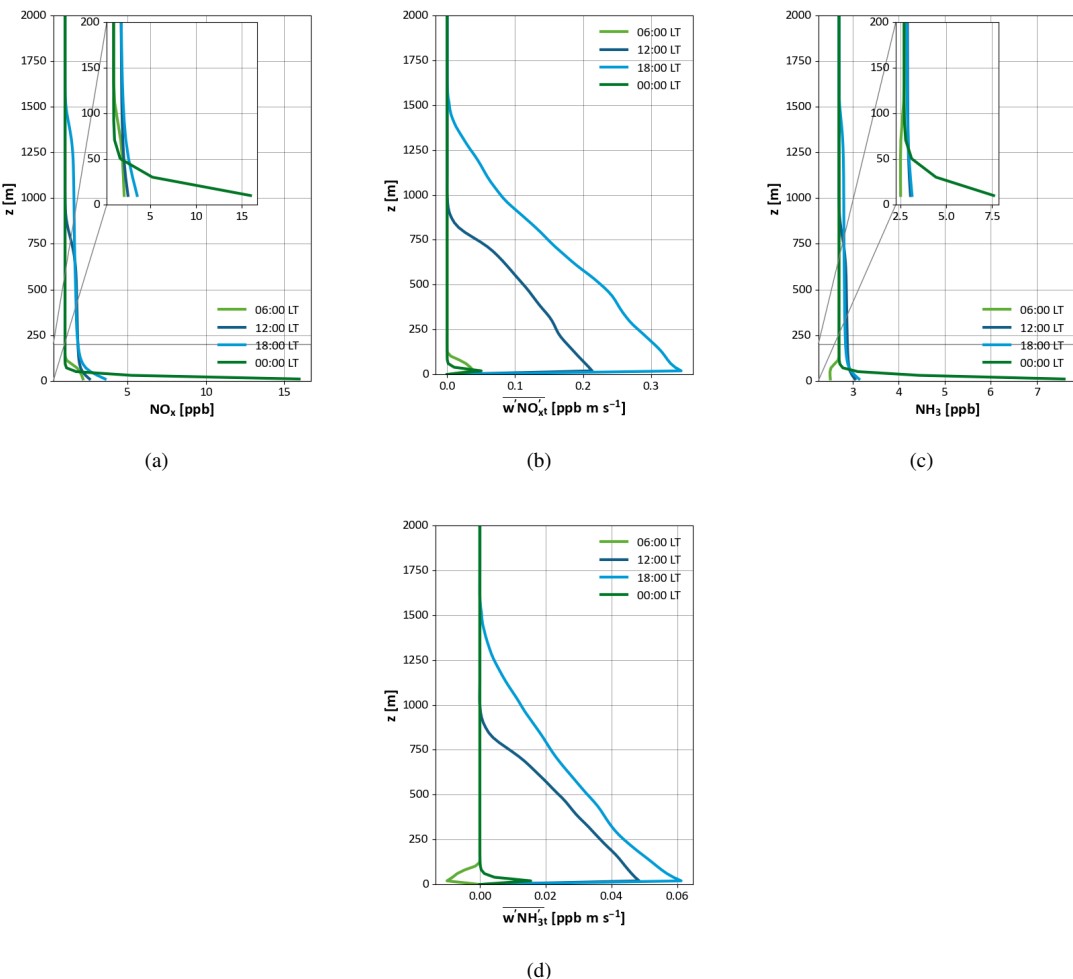

**Figure 7.** Vertical profiles at different times of $NO_x$ (a) and $NH_3$ concentrations (c), the total turbulent fluxes of both species (b & d). In (a) and (b) a zoom-in on the lowest 200 m is shown inset.

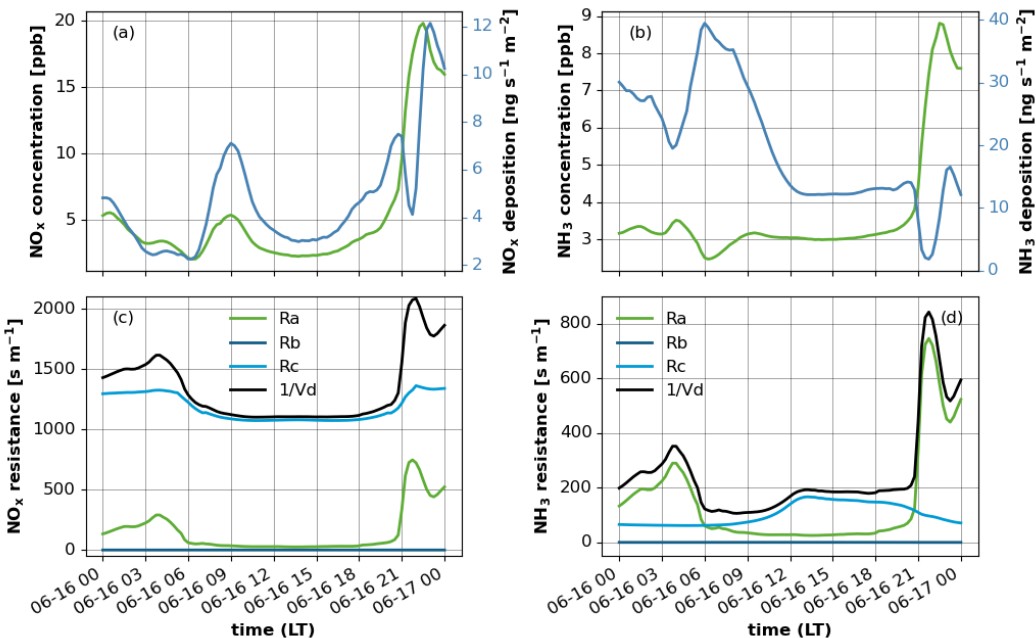

**Figure 8.** Domain averaged diurnal cycle of concentration and deposition flux of $NO_x$ (a) and $NH_3$ (b), and of the reciprocal deposition velocity ($1/V_d$) and the aerodynamic ($R_a$), the quasi-laminar ($R_b$) and canopy ($R_c$) resistance for $NO_x$ (c) and $NH_3$ (d).

in the present simulations, we imposed a constant concentration on the boundaries (1 ppb, amounting to approx. 1.3 µg m$^{-3}$), which ignores the presence of sources outside the domain, but does provide a realistic background concentration for $NO_x$. DALES is currently being extended by implementing the possibility to nest a calculation in a bigger domain, enabling the use of boundary conditions originating from a parent domain from DALES or a regional-scale atmospheric model (Liqui Lung et al., 2024). This will enable setting realistic boundary conditions with a more accurate background concentration, and incorporating the effect of plumes generated outside the domain.

The concentration difference between the lowest layer of the atmosphere and the compensation point (which is 0 for $NO_x$) is the driving force of the deposition flux (Figure 9b). The deposition velocity (Figure 9c) is the conversion factor between this concentration difference and the deposition flux. Note that we treated $NO_x$ as $NO_2$ in our deposition calculations, since in the atmosphere, NO emissions are quickly chemically converted to $NO_2$. Due to its LU dependence, the deposition velocity shows a clear footprint of the LU map. Grassland and open fields show low values (0.04 cm s$^{-1}$) for deposition velocity, increasing a little for urban areas (0.06 cm s$^{-1}$) and reaching maximum values over forests (0.08 cm s$^{-1}$). For $NO_x$ these differences are mainly due to the difference in roughness length between the LU types.

The $NH_3$ concentration field at 08:00 LT mainly reflects the emissions from agriculture, and hence we find concentration maxima near farm locations at the northern and western edges of the domain (Figure 10a). Also the areas downwind of the highways show elevated concentrations. The atmospheric lifetime of a primary air pollutant and thus its traveling distance is

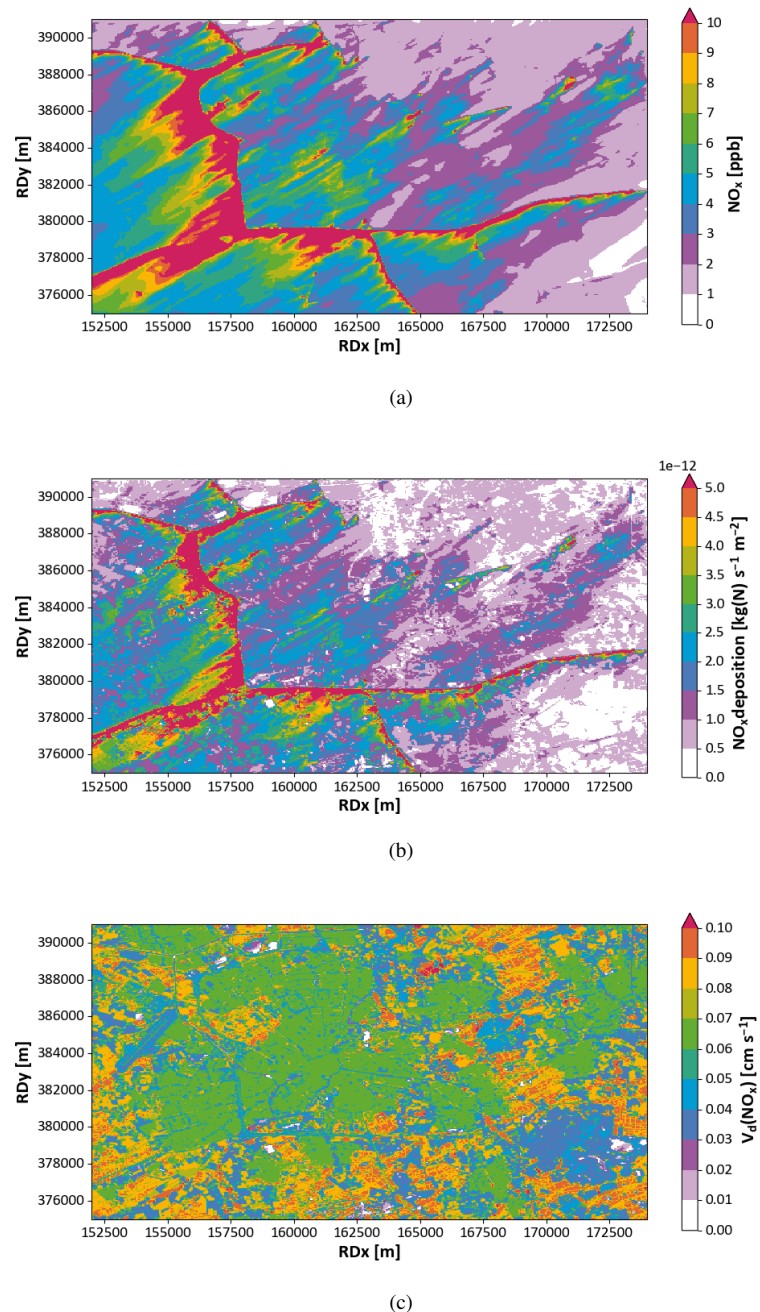

**Figure 9.** Distribution of $NO_x$ concentration in the lowest layer of the model (z=10 m) (a), distribution of the deposition flux of $NO_x$ (b) and distribution of (c) the deposition velocity of $NO_x$ over the domain at 08:00 LT. Note that $NO_x$ is deposited as $NO_2$.

governed by an intricate balance between height of emission, advection and turbulent transport, as well as the removal rates by dry and wet deposition. Specifically for reactive and water soluble pollutants emitted close to ground level, like $NH_3$, the travel distance is shortened by deposition fluxes. This means that species are allowed to travel further over land with a low deposition flux, like grassland and urban areas. Forests are effective to shorten travel distances.

The modeled $NH_3$ deposition fluxes are about an order of magnitude larger than those of $NO_x$ (Figure 10b). Similarly to $NO_x$, the influence of LU is clearly visible in the deposition flux $NH_3$: the smallest fluxes are calculated in urban areas and water bodies, whereas the largest deposition fluxes are found over forest areas. The striking features are modeled near the transition from agricultural land (pastures and cropland) to forest in the northeastern part of the domain (indicated by the box in Fig.10b): here, plumes enriched with $NH_3$ are advected from a smooth to a rough surface. Over the rough surface, turbulent exchange is enhanced, which leads to rapid deposition of $NH_3$. Further into the forest the deposition fluxes become smaller than near the transition. This shows that a high resolution, turbulence resolving model like DALES is uniquely suited to simulate this kind of small-scale features, in which the interactions between the turbulent transport and the change in local LU together determine the deposition flux.

A limitation of our study is the fact that we ignore possible effects of chemistry and aerosol formation on the deposition of $NO_x$ and $NH_3$. Under atmospheric conditions, $NO_x$ will be partially converted into $HNO_3$ which deposits more readily than its precursor and which can be neutralized with $NH_3$ to form ammonium nitrate aerosol. DALES has been used before to study the effects of turbulent mixing on the phase transition of ammonium nitrate. Aan de Brugh et al. (2013) found that aerosol-poor air is transported upward from the surface and aerosol-rich is transported from high altitudes downward, since equilibrium between the gas and aerosol phase is not instantaneous. Further, Barbaro et al. (2015) found large deposition velocities for nitrate due to outgassing near the surface. Finally, a study with another LES code concluded that the effective Henry's law constant is a critical factor for parameterization of dry deposition of gas-phase species in a street canyon (Lin et al., 2024). Taking into account the effects of gas-phase chemistry and gas-aerosol partitioning will have considerable impacts on the calculated deposition fluxes. We aim to cover the effects of a full coupling between emissions, chemistry, partitioning and turbulent transport in a follow-up study.

# 4 Evaluation against observations at Cabauw

To ensure the validity of the deposition fluxes calculated by DALES, an evaluation against measurements is urgently needed. However, the possibilities for such an evaluation are limited by the availability of reliable flux measurements (Wintjen et al., 2022). In this section, we therefore provide a limited evaluation for a single day. Because no observations of deposition fluxes are available for Eindhoven, we here evaluate DALES against observations at the Cabauw experimental site in The Netherlands (51.97 °N, 4.93 °E). This site is at an almost flat terrain consisting primarily of grassland. Input data for DALES calculations was acquired in the same way as for the Eindhoven case (Sections 2.4 and 2.5). The concentration and deposition flux of $NH_3$ were calculated on September 25, 2021 (dry day, SSW wind turning E, 2-4 m s$^{-1}$, 15-21°C, 70% RH during the day, 90% at night). Measurements of $NH_3$ concentration and deposition flux for this day were acquired, as part of the RITA-2021 campaign,

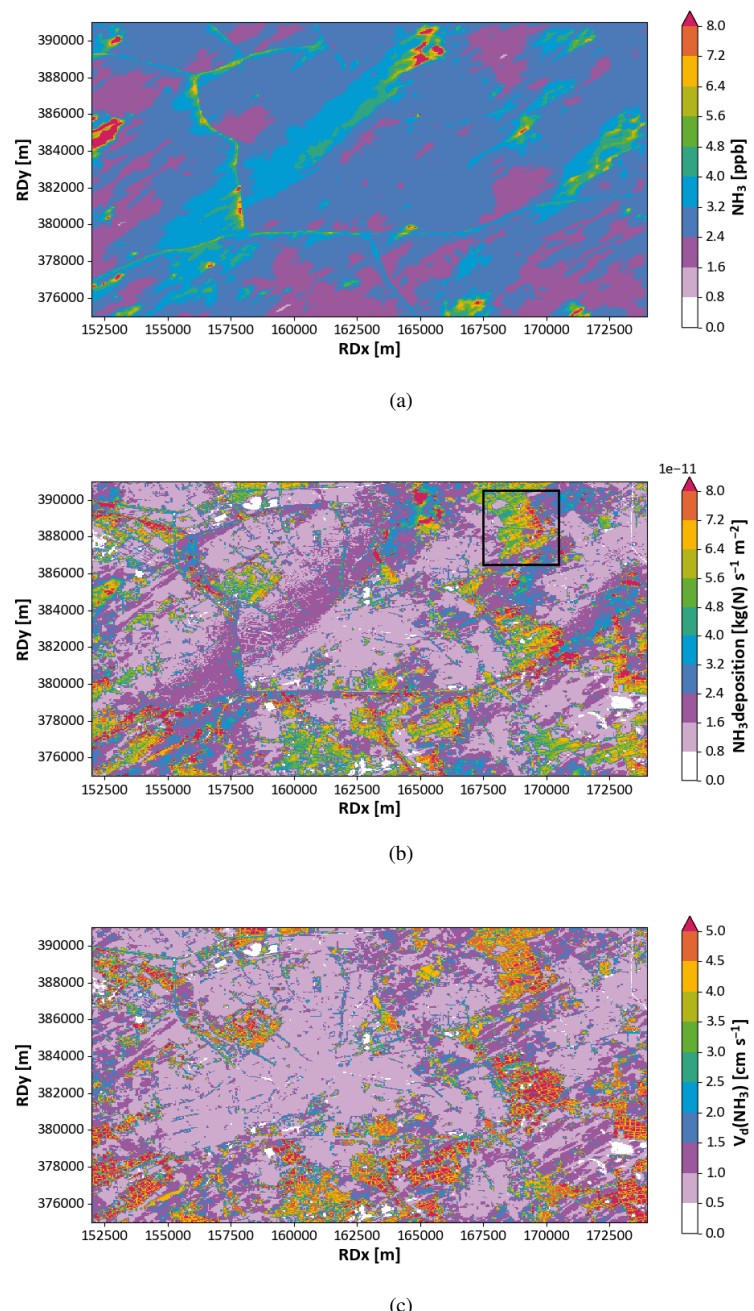

**Figure 10.** Distribution of $NH_3$ concentration in the lowest layer of the model (z=10 m) (a), distribution of the deposition flux of $NH_3$ (b) and distribution of (c) the deposition velocity of $NH_3$ over the domain at 08:00 LT.

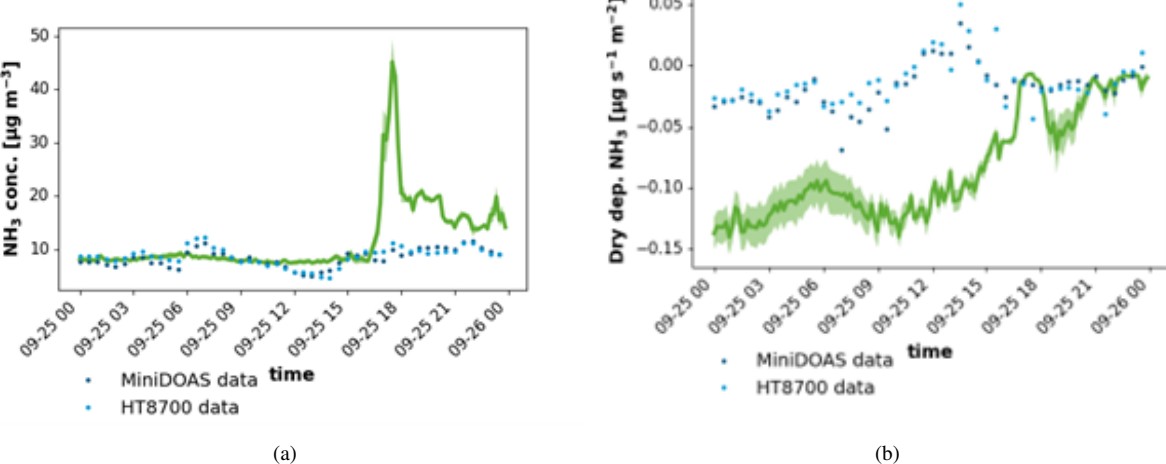

(a)                                                                                          (b)

**Figure 11.** Modeled and measured concentration (a) and deposition flux (b) at Cabauw compared to measurement data by Mini-DOAS and HT8700. DALES results are shown by the green line with the shades indicating the standard deviation of the DALES output for grid cells within a radius of 100 m around the data sampling location.

by a Healthy Photon HT8700E open path ammonia analyzer (eddy covariance flux analyzer) and the RIVM-miniDOAS 2.2D (differential optical absorption spectroscopy) instrument (Swart et al., 2023).

Figure 11 shows the $NH_3$ concentration and deposition flux during the day. The concentration was predicted around 9 $\mu g\,m^{-3}$, which is comparable to the measured concentrations. The predicted concentration peak in the evening does not reflect the measured data, because of the transition from an instable to a stable boundary layer that is not yet covered well by DALES. The deposition flux is overestimated by the model in the morning and afternoon, but it settles to a value around the measurement data after the wind subsided and its direction changed in the evening. DEPAC is known to overestimate

deposition fluxes particularly during early morning hours (Jongenelen et al., 2025). One cause for this over-prediction can be the overestimation of LAI in DEPAC. The value for grassland is estimated at 3 on September 25, 2021, whereas the MODIS satellite LAI measurements show values between 1 and 2 at Cabauw in that period. A further evaluation over other land use types would be commendable, but that depends on the availability of reliable flux measurements of reactive nitrogen, which are sparse (Wintjen et al., 2022).

**5   Conclusions**

Studying the fate of reactive nitrogen compounds in a complex landscape, while accounting for the interplay between dispersion, chemistry and deposition processes, requires a detailed model setup at very high spatial and temporal resolution. We have taken the first steps in preparing the Dutch Atmospheric Large Eddy Simulation (DALES) model for such realistic applications.

As a first step we successfully integrated the dry deposition module based on DEPAC in DALES, supported by a flexible land use definition. Also, we provided high-resolution emissions of $NO_x$ and $NH_3$ for the model system.

In a case study for the Eindhoven region we found that DALES is able to reproduce the main features of the boundary layer development and diurnal cycle of local meteorology well, with the exception of the evening transition. DALES calculates the dispersion and deposition of $NO_x$ and $NH_3$ in great spatial detail, clearly showing the influence of local land use patterns on removal efficiencies and mixing characteristics. This shows the promise of the model for deposition studies in complex landscapes. To further develop the system for realistic applications we are working on the detailing of the emissions on the required scale, the integration of gas phase chemistry, inorganic aerosol formation and the accommodation of open boundary conditions. In particular regarding emissions, the adoption of a dynamic scheme will reduce the uncertainty of emission fluxes and local concentration estimates.

*Code availability.* DALES is released under the GPLv3 license and it is made available to the general public on the Github repository found at GitHub - dalesteam/dales: Dutch Atmospheric Large-Eddy Simulation model. The calculations in this paper were performed on the basis of branch '4.4_Ruisdael_deposition' (https://github.com/dalesteam/dales/tree/4.4_Ruisdael_deposition). The exact version of the model used to produce the results used in this paper is archived on Zenodo at https://doi.org/10.5281/zenodo.15547107 (Geers et al., 2025).

## Appendix A: LSM and DEPAC parameters

**Table A1.** Parameters for the HTESSEL land surface model (ECMWF, 2021; Balsamo et al., 2009).

| LU type parameter | aqu | ara | brn | crp | fbd | fce | grs | sem | urb |
|---|---|---|---|---|---|---|---|---|---|
| $f_{min}$ (-) | 0.0 | 0.01 | 0.0 | 0.01 | 0.1 | 0.1 | 0.01 | 0.04 | 0.0 |
| $LAI$ ($m^2 m^{-2}$) | 0.0 | 3.0 | 2.0 | 1.5 | 5.0 | 5.0 | 2.0 | 0.5 | 2.0 |
| $R_{s,min}$ ($sm^{-1}$) | $1.0 \cdot 10^9$ | 180.0 | 100.0 | 225.0 | 250.0 | 250.0 | 100.0 | 150.0 | 100.0 |
| $z_{0,h}$ (m) | 0.001 | 0.005 | 0.002 | 0.0025 | 2.0 | 2.0 | 0.002 | 0.0017 | 1.0 |
| $z_{0,m}$ (m) | 0.1 | 0.5 | 0.2 | 0.25 | 2.0 | 2.0 | 0.2 | 0.17 | 1.0 |

**Table A2.** Parameters for the deposition model (Van Zanten et al., 2010)

| LU type parameter | aqu | ara | brn | crp | fbd | fce | grs | sem | urb |
|---|---|---|---|---|---|---|---|---|---|
| $b$ ($m^{-1}$) | 0.0 | 14.0 | 0.0 | 14.0 | 14.0 | 14.0 | 0.0 | 14.0 | 0.0 |
| $h$ (m) | 0.0 | 1.0 | 0.0 | 2.5 | 20.0 | 20.0 | 0.0 | 1.0 | 0.0 |
| $a_{SAI}$ (-) | 0.0 | 1.0 | 0.0 | 1.0 | 1.0 | 1.0 | 1.0 | 1.0 | 0.0 |
| $b_{SAI}$ (-) | 0.0 | 1.5 | 0.0 | 0.5 | 1.0 | 1.0 | 0.0 | 0.0 | 0.0 |
| $\Gamma_{soil}$ (-) | 430.0 | 0.0 | 0.0 | 0.0 | 0.0 | 0.0 | 0.0 | 0.0 | 0.0 |
| $\Gamma_s$ (-) | 0.0 | 362.0 | 0.0 | 362.0 | 362.0 | 362.0 | 362.0 | 362.0 | 0.0 |
| $G_s^{max}$ ($\cdot 10^{-3}$ $ms^{-1}$) | 0.0 | 7.317 | 0.0 | 7.317 | 3.659 | 3.415 | 6.585 | 1.024 | 0.0 |

**Table A3.** Model parameters for soil resistance $R_{soil}$ (s m$^{-1}$) (based on Erisman et al., 1994).

| LU type species | aqu | ara | brn | crp | fbd | fce | grs | sem | urb |
|---|---|---|---|---|---|---|---|---|---|
| $NH_3$ | 10 | 100 | 100 | 100 | 100 | 100 | 100 | 100 | 100 |
| $NO_2$ | 2000 | 1000 | 1000 | 1000 | 1000 | 1000 | 1000 | 1000 | 1000 |
| $NO$ | 2000 | $\infty$ | 2000 | $\infty$ | $\infty$ | $\infty$ | $\infty$ | $\infty$ | 1000 |

## Appendix B: Land use translation

**Table B1.** Translation table of TOP10NL land use types (in Dutch) to DEPAC land use types

| DEPAC LU type | DEPAC short name | TOP10NL LU types |
|---|---|---|
| Aquatic | aqu | Waterloop, Meer/plas, Zee, Water droogvallend, Water droogvallend (LAT), Aanlegstijger |
| Arable land | ara | Akkerland |
| Barren land | brn | Zand, Weg onverhard, Braakliggend |
| Permanent crops | crp | Boomkwekerij, Fruitkwekerij, Boomgaard |
| Broadleaf deciduous forest | fbd | Loofbos, Griend, Populieren, Dodenakker met bos |
| Coniferous evergreen forest | fce | Naaldbos, Gemengd bos |
| Grassland | grs | Grasland |
| Semi-natural | sem | Heide, Duin, Dodenakker |
| Urban | urb | Weg verhard, Weg half verhard, Weg onbekend, Spoorbaanlichaam, Basaltblokken/steenglooiing, Bebouwd gebied, Overig |

## Appendix C: Boundary layer height diagnostics

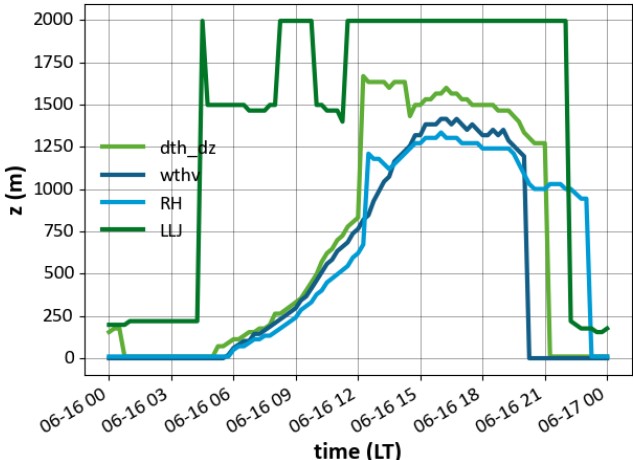

**Figure C1.** Boundary layer height from DALES as diagnosed by 4 different methods: heights of the maximum $\Delta\theta_v/\Delta h$, minimum value of $\overline{w'\theta_v'}$, maximum value of RH and low level jet, respectively.

*Author contributions.* LG implemented the extended DEPAC module in DALES and wrote the manuscript. RJ contributed to the model implementation, performed the model calculations, and wrote the manuscript. GT performed model calculations. JV-GdA provided advice on the DALES simulations and the interpretation of the results. MS coordinated the model development and wrote the manuscript. All co-authors commented on the manuscript draft.

*Competing interests.* The authors declare no competing interests.

*Acknowledgements.* We acknowledge Bart van Stratum for his work on the land surface module, and Ingrid Super and Stijn Dellaert for their work on the emission downscaling tool. We acknowledge Roy Wichink Kruit and Margreet van Zanten for their collaboration and support on DEPAC.

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
