# Peer review of "Implementation of a dry deposition module (DEPAC v3.11\_ext) in a large eddy simulation code (DALES v4.4)"

_EGUsphere, 2025_

## Author Comment (AC1)

Dear Reviewer,

Thank you for your review of our manuscript. We appreciate your valuable comments and suggestions and the opportunity to address your concerns and provide clarifications. Below, we respond to each of the major and minor comments raised. Our responses are in blue font.

This study implemented a deposition module into a large-eddy simulation code and presented a case study on the dispersion and deposition of $NO_x$ and $NH_3$. However, due to the lack of model sensitivity analysis, it remains unclear whether the implemented model accurately reproduces gas-phase deposition. Below are several major concerns:

Before applying the developed deposition module within DALES, a theoretical analysis should be conducted to examine the relationship between deposition velocity and the dry deposition parameters of each gas species in DEPAC. These relationships should then be compared with existing literature to validate the correctness and reliability of the DEPAC implementation.

We understand the need for an evaluation of the DEPAC module, as there are many factors that can influence the deposition velocity of a gas phase species. However, we think that a full theoretical analysis for each gas-phase species is beyond the scope of this paper. Besides, DEPAC has been applied as a standalone deposition module before and has been extensively evaluated against observations. We will refer to the relevant literature on the DEPAC module in Section 2.3.

Added new subsection (P6L152) "Calculation of resistances"

Changed (P6L152): "In this work, the DEPAC deposition module is used, which is an implementation of the resistance model."

To: "The DEPAC deposition module applied in this work is an implementation of the resistance model."

Added (P6L154):

"The DEPAC module is a well-established module for dry deposition calculations. It is used as a dry deposition module in the air quality models LOTOS-EUROS (Manders et al., 2017, 2022) and OPS (Sauter et al., 2020). A theoretical analysis of the sensitivity of DEPAC to several of its input parameters is given by (Van Zanten et al., 2010). Further, it has been evaluated against observed deposition fluxes over forests (Melman et al., 2025; Wintjen et al., 2022) and dune ecosystems (Jongenelen et al., 2025; Vendel et al., 2023). These analyses have shown that the parameterizations of compensation point and the external resistance ($R_w$)

contribute most to uncertainties in calculated deposition (and emission) fluxes of $NH_3$."

The condensation and evaporation processes of $HNO_3$ and $NH_3$ can have effects comparable to their dry deposition, and different deposition schemes may even reverse the gas-particle partitioning (Lin et al., 2024). However, aerosol dynamics are not considered in this study, which introduces substantial uncertainty into the case study results. The exclusion of this key mechanism undermines confidence in the conclusions drawn from the simulations.

We agree that the condensation and evaporation processes of $HNO_3$ and $NH_3$ are critical factors that can influence dry deposition of these species. In the revised manuscript, we will incorporate a discussion on the potential effects of these processes and acknowledge the limitations of not including aerosol dynamics in our current study. Implementing aerosol dynamics in DALES will be addressed in future work.

Added text, P19L425:

"A limitation of our study is the fact that we ignore possible effects of chemistry and aerosol formation on the deposition of $NO_x$ and $NH_3$. Under atmospheric conditions, $NO_x$ will be partially converted into $HNO_3$ which deposits more readily than its precursor and which can be neutralized with $NH_3$ to form ammonium nitrate aerosol. DALES has been used before to study the effects of turbulent mixing on the phase transition of ammonium nitrate. Aan de Brugh et al. (2013) found that aerosol-poor air is transported upward from the surface and aerosol-rich is transported from high altitudes downward, since equilibrium between the gas and aerosol phase is not instantaneous. Further, Barbaro et al. (2015) found large deposition velocities for nitrate due to outgassing near the surface. Finally, a study with another LES code concluded that the effective Henry's law constant is a critical factor for parameterization of dry deposition of gas-phase species in a street canyon (Lin et al., 2024).

Taking into account the effects of gas-phase chemistry and gas-aerosol partitioning will have considerable impacts on the calculated deposition fluxes. We aim to cover the effects of a full coupling between emissions, chemistry, partitioning and turbulent transport in a follow-up study."

The spatial and temporal distributions of Ra, Rb, Rc, deposition velocity, and deposition flux during the simulation period should be provided and discussed. This would allow for a more comprehensive understanding of the deposition process. Furthermore, it is important to clarify which of the resistances (Ra, Rb, or Rc) is the dominant factor under the simulated conditions.

We have added temporal distributions of $R_a$, $R_b$, $R_c$, deposition velocity, and deposition flux during the simulation period to the revised manuscript, in addition to the maps of deposition flux and velocity that were already included (Figures 8 and 9). Since the new Figure 8a and 8b show the concentration time series, these

[Figure]

Figure 1(Fig. 8 in the revised MS): diurnal cycle of concentration and deposition flux of $NO_x$ (a) and $NH_3$ (b), and of the deposition velocities and the aerodynamic ($R_a$), the quasi-laminar ($R_b$) and canopy ($R_c$) resistance for $NO_x$ (c) and $NH_3$ (d).

were removed from the current Figure 7e.

P16L381:

"The diurnal cycle of $NO_x$ shows a strong relationship between the concentration and the flux (Figure 8a), indicating that the deposition flux is mainly driven by the concentration gradient between the atmosphere and the surface. It also shows that during most of the day, the canopy resistance ($R_c$) is the most important factor that drives the deposition velocity ($V_d$; Figure 8c). For $NO_x$, the $R_c$ mostly follows the stomatal resistance, as shown by the decreasing $R_c$ during daytime. Only during nighttime, the aerodynamic resistance ($R_a$) plays a significant role in determining the deposition velocity.

For $NH_3$, a different picture emerges (Figure 8b). During nighttime (between 00:00 and 06:00 LT), the NH3 concentration and deposition are coupled, but during the morning (between 06:00 and 10:00 LT), the concentration increases while the deposition flux decreases. This is driven by an increasing $R_c$ (Figure 8d). For $NH_3$, the uptake on wet external surfaces is the most important contribution to the canopy

uptake, and this pathway decreases when dew on the leaves evaporates. During daytime (10:00 to 20:00 LT), concentration and flux remain relatively constant. In the evening (between 20:00 and 00:00 LT), the deposition flux increases due the increased concentration, while the $R_c$ decreases again due to dew formation. The $V_d$ is dampened by the rising $R_a$.

For both species, $R_b$ only plays a minor role in determining the deposition velocities during the whole day."

Minor comments:

Line 196:' In addition, a sensitivity analysis in a related project pointed out that the effect of switching between dry and wet land on the deposition fluxes of NH3 is not very strong.'

Please provide a citation or additional evidence to support this statement. Without substantiation, the conclusion may appear speculative.

This statement was found to be erroneous on further investigation of the related project. The deposition velocity for NO and $NO_2$ only differ by approximately 12% between a completely wet domain vs a dry surface (save for some waterways). The difference for $NH_3$ is much larger. Note that this is a worst case scenario, where the complete surface is saturated with water. In that case, $NH_3$ deposition is controlled by the high solubility of $NH_3$ in water.

We did want to be sure that working only with the equations for dry would be justified. KNMI weather data at the location of Eindhoven Airport showed that the date of our simulations was preceded by a prolonged period of dry conditions (11 days without any precipitation). We have added a line in the discussion of the Eindhoven case study stating this and the statement on the related project was deleted.

In P8L196, we replaced:

"Here, we circumvented this problem by selecting dry days in a period with little or no rain. In addition, a sensitivity analysis in a related project pointed out that the effect of switching between dry and wet land on the deposition fluxes of $NH_3$ is not very strong."

By:

"In our case study, we circumvented this problem, since the KNMI weather data at the location of Eindhoven Airport showed that the date of our simulations was preceded by a prolonged period of dry conditions (11 days without any precipitation)."

References

Aan de Brugh, J. M. J., Ouwersloot, H. G., Vilà-Guerau de Arellano, J., and Krol, M. C.: A large-eddy simulation of the phase transition of ammonium nitrate in a convective boundary layer, Journal of Geophysical Research: Atmospheres, 118, 826–836, https://doi.org/10.1002/jgrd.50161, 2013.

Barbaro, E., Krol, M. C., and Vilà-Guerau De Arellano, J.: Numerical simulation of the interaction between ammonium nitrate aerosol and convective boundary-layer dynamics, Atmospheric Environment, 105, 202–211, https://doi.org/10.1016/j.atmosenv.2015.01.048, 2015.

Jongenelen, T., van Zanten, M., Dammers, E., Wichink Kruit, R., Hensen, A., Geers, L., and Erisman, J. W.: Validation and uncertainty quantification of three state-of-the-art ammonia surface exchange schemes using NH3 flux measurements in a dune ecosystem, Atmospheric Chemistry and Physics, 25, 4943–4963, https://doi.org/10.5194/acp-25-4943-2025, 2025.

Lin, C., Ooka, R., Kikumoto, H., Kim, Y., Zhang, Y., Flageul, C., and Sartelet, K.: Impact of gas dry deposition parameterization on secondary particle formation in an urban canyon, Atmospheric Environment, 333, 120633, https://doi.org/10.1016/j.atmosenv.2024.120633, 2024.

Manders, A. M. M., Builtjes, P. J. H., Curier, L., Denier van der Gon, H. A. C., Hendriks, C., Jonkers, S., Kranenburg, R., Kuenen, J. J. P., Segers, A. J., Timmermans, R. M. A., Visschedijk, A. J. H., Wichink Kruit, R. J., van Pul, W. A. J., Sauter, F. J., van der Swaluw, E., Swart, D. P. J., Douros, J., Eskes, H., van Meijgaard, E., van Ulft, B., van Velthoven, P., Banzhaf, S., Mues, A. C., Stern, R., Fu, G., Lu, S., Heemink, A., van Velzen, N., and Schaap, M.: Curriculum vitae of the LOTOS–EUROS (v2.0) chemistry transport model, Geosci. Model Dev., 10, 4145–4173, https://doi.org/10.5194/gmd-10-4145-2017, 2017.

Manders, A. M. M., Segers, A. J., and Jonkers, S.: LOTOS-EUROS v2.2.003 Reference Guide, 2022.

Melman, E. A., Rutledge-Jonker, S., Frumau, K. F. A., Hensen, A., van Pul, W. A. J., Stolk, A. P., Wichink Kruit, R. J., and van Zanten, M. C.: Measurements and model results of a two-year dataset of ammonia exchange over a coniferous forest in the Netherlands, Atmospheric Environment, 344, 120976, https://doi.org/10.1016/j.atmosenv.2024.120976, 2025.

Sauter, F. J., Sterk, H. A. M., van der Swaluw, E., Wichink Kruit, R. J., de Vries, W., and Van Pul, W. A. J.: The OPS-model; Description of OPS 5.0.0.0, 2020.

Van Zanten, M. C., Wichink Kruit, R. J., van Jaarsveld, H. A., and van Pul, W. A. J.: Description of the DEPAC module : Dry deposition modelling with DEPAC_GCN2010, Rijksinstituut voor Volksgezondheid en Milieu RIVM, 2010.

Vendel, K. J. A., Wichink Kruit, R. J., Blom, M., van den Bulk, P., van Egmond, B., Frumau, A., Rutledge-Jonker, S., Hensen, A., and van Zanten, M. C.: Dry deposition of ammonia in a coastal dune area: Measurements and modeling, Atmospheric Environment, 298, 119596, https://doi.org/10.1016/j.atmosenv.2023.119596, 2023.

Wintjen, P., Schrader, F., Schaap, M., Beudert, B., Kranenburg, R., and Brümmer, C.: Forest–atmosphere exchange of reactive nitrogen in a remote region – Part II: Modeling annual budgets, Biogeosciences, 19, 5287–5311, https://doi.org/10.5194/bg-19-5287-2022, 2022.

---

## Author Comment (AC2)

Dear Dr. Suter,

Thank you for your detailed review and your constructive comments on our manuscript. Below, we respond to each of the major and minor comments raised in blue font.

The authors describe the extension of an atmospheric LES (DALES) with a dry deposition module (DEPAC), with the aim to better spatially resolve deposition of ammonia and $NO_2$. The manuscript presents the development of a suitable tool to study deposition on smaller scale, relevant for very heterogeneous terrain and areas of particular interest, such as water conservation or nature preservation zones. As such, the presented work clearly is of sufficient interest to warrant publication in geoscientific model development.

However, while the methods are described theoretically, their actual implementation into the model remain somewhat unclear from the manuscript. The information can be obtained from the provided source code, of course. But this seems unnecessarily cumbersome. This is particularly true, since DEPAC is already described in the referenced literature. The authors should mention some aspects of implementation that are helpful to readers who might want to attempt a similar undertaking (e.g. are the deposition rates added to the tendencies? How is R=infinity realised? Are molecular or sub-grid diffusivities being used to calculate Rb).

We acknowledge that the theoretical description of the methods may not fully clarify their implementation into the model. To address this, we have added a section in the revised manuscript detailing key aspects of the DEPAC implementation. This includes how deposition rates are added to the tendencies, the realization of R=inf, and the use of molecular or sub-grid diffusivities to calculate Rb.

Added new section for implementation in DALES, P9L247:

"2.3.3 Implementation in DALES

In the main driver of DALES, for each time step (i.e. every few seconds), the dry deposition routine is called in a section of surface routines, after the radiation terms are calculated and before advection and diffusion terms are calculated. First, the necessary parameters like LAI and SAI are calculated based on values from a parameter table. Then the deposition budget is calculated for each depositing tracer is by a call to the DryDepos_Gas routine in the dry deposition module. Finally, in a loop over all depositing tracers the deposition budget for each tracer is added to the tracer tendency array in all surface cells."

For the realization of R=inf, we added at P9L239:

"In case one of the resistances is irrelevant (e.g., in case of stomatal resistance for aqueous surfaces), the resistance value is set to -9999, a special value to indicate that the resistance is not to be considered in the calculation of the deposition velocity."

Regarding Rb calculations, P7L168:

"In the calculation of Rb, the molecular diffusivity and viscosity are used."

The authors could not convince me that the modelled boundary layer height after sunset is not a problem. I encourage them to attempt to improve this (could a different sub-grid model or forcing help?). In the current state the inclusion of midnight (00:00 LT) into the analysis is somewhat moot.

We understand the reviewer's concern regarding the modelled boundary layer height after sunset, and we do acknowledge that it is a problem. Our intention was to be honest about the current capability of DALES to simulate the transition from an unstable to a stable boundary layer.

That is the reason why we have performed and described (lines 344-346) an additional simulation in which the horizontal resolution was decreased to 25x25 m$^2$, which did not solve the evening transition issue. In the revised version, we included results from this simulation. Note that DALES is able to reproduce the typical potential temperature profile of a stable nocturnal boundary layer, see the profile in Figure 5a at 00:00.

The revised Figure 5d shows the boundary layer development of both the default simulation (50 x 50 m$^2$ resolution) and the additional simulation at 25 x 25 m$^2$. The strong collapse of the boundary layer is present in both simulations, although it occurs about 30 minutes earlier in the 25 x 25 m$^2$ simulation. We include this figure in the revised paper. In addition, the time series of temperature, humidity, and wind speed hardly show any impact of the resolution. We will not include this figure in the revised paper.

As subgrid model, DALES uses the SFS-TKE *e* scheme (Deardorff, 1980) by default. The only available alternative in v4.4 is the Smagorinsky subgrid model, which also has known limitations for the stable boundary layer (Beare et al., 2006).

As recently as last week, the subgrid scheme by Dai et al. (2021) has been implemented in DALES (https://github.com/dalesteam/dales/pull/185). It allows for the simulation of stable nocturnal boundary layers with coarse resolution. We aim to apply this scheme in follow-up studies that cover a full diurnal cycle.

We added (P14L347): "In addition to using a finer resolution, applying a different sub grid model may help to improve the simulation of the transition to a stable

boundary layer. For this study, we applied the SFS-TKE *e* scheme (Deardorff, 1980) which is the default in DALES. In future studies, the use of a sub-grid model that for the simulation of stable nocturnal boundary layers with coarse resolution (Dai et al., 2021) will be explored."

[Figure]

*Figure 1: time series of temperature, humidity and wind for the default and the 25 x25 m² simulations*

[Figure]

*Figure 2 (revised figure 5d): boundary layer height and backscatter for the control run (DALES) and the high resolution run (25 x 25 m)*

There are several inaccuracies and inconsistencies about units, variable names etc. and several plots can be improved. These have been pointed out as individual comments.

Thank you for pointing these out. We have carefully reviewed the manuscript to correct inaccuracies and inconsistencies related to units, variable names and plots.

I think the copyright statement in le_drydepos_gas_depac.f90 does not match GPLv3.

We have checked and updated the copyright statement for the deposition model. In correspondence with the copyright holder of DEPAC (RIVM), an arrangement was made to enable use of the code in DALES. Now the code is available under GPLv3, but the use of the name DEPAC is restricted, since the model has been extended and adapted since first publication under that name. To prevent further confusion, we've eliminated the use of the name DEPAC from the routines, while the original comments with explanations about the model and the changelog remain unchanged. Also, we have updated the title of the MS to reflect this: 'DALES v3.11' will be changed to 'DALES v3.11_ext' (DALES v3.11 with extensions).

Lastly, a model validation against measurements would contribute significantly to the article. The authors make clear, that the model in the current state does not

contain a chemistry module and is still under development, which unfortunately makes this task currently difficult. Maybe a weak comparison to NOx measurements is already possible?

We've added Section 4, which shortly shows a comparison between a DALES simulation and measured data of the concentration and deposition flux of $NH_3$ at Cabauw, a well-established experimental site in The Netherlands. These data provide a first insight in the validity of the modeled concentrations and deposition fluxes. This is what we added (P21L425):

"4. Evaluation against observations at Cabauw

To ensure the validity of the deposition fluxes calculated by DALES, an evaluation against measurements is urgently needed. However, the possibilities for such an evaluation are limited by the availability of reliable flux measurements (Wintjen et al., 2022). In this section, we therefore provide a limited evaluation for a single day. Because no observations of deposition fluxes are available for Eindhoven, we here evaluate DALES against observations at the Cabauw experimental site in The Netherlands (51.97° N, 4.93° E). This site is at an almost flat terrain consisting primarily of grassland. Input data for DALES calculations was acquired in the same way as for the Eindhoven case (Sections 2.4 and 2.5).

The concentration and deposition flux of $NH_3$ were calculated on September 25, 2021 (dry day, SSW wind turning E, 2-4 m/s, 15-21° C, 70% RH during the day, 90% at night). Measurements of $NH_3$ concentration and deposition flux for this day, were acquired as part of the RITA-2021 campaign by a Healthy Photon HT8700E open path ammonia analyzer (eddy covariance flux analyzer) and the RIVM-miniDOAS 2.2D instrument (differential optical absorption spectroscopy) (Swart et al., 2023).

[Figure]

[Figure]

Fig 3 (Fig 11 in the revised MS): Modeled and measured concentration (a) and deposition flux (b) at Cabauw compared to measurement data by Mini-DOAS and HT8700. DALES results are shown by the green line with the shades indicating the

standard deviation of the DALES output for grid cells within a radius of 100 m around the data sampling location.

Figure 11 shows the $NH_3$ concentration and deposition flux during the day. The concentration was predicted around 9 μg/m$^3$, which is comparable to the measured concentrations. The predicted concentration peak in the evening does not reflect the measured data, because of the transition from an instable to a stable boundary layer that is not yet covered well by DALES.

The deposition flux is overestimated by the model in the morning and afternoon, but it settles to a value around the measurement data after the wind subsided and its direction changed in the evening. DEPAC is known to overestimate deposition fluxes particularly during early morning hours (Jongenelen et al., 2025). One cause for this overprediction can be the overestimation of LAI in DEPAC. The value for grassland is estimated at 3 on September 25, 2021, whereas the MODIS satellite LAI measurements show values between 1 and 2 at Cabauw in that period.

A further evaluation over other land use types would be commendable, but that depends on the availability of reliable flux measurements of reactive nitrogen, which are sparse (Wintjen et al., 2022)."

I strongly encourage a proper validation as soon as possible.

We fully agree with the reviewer that a proper validation of the simulated tracer concentrations and deposition fluxes is of high value to establish the accuracy of the coupled model. Therefore, we have included an evaluation over a (predominantly) grassland area in the revised manuscript (see the previous point).

Other comments:

P3L76 Why is only the OH marked as a radical?

That's an oversight, HO2 is also a radical. This was corrected in the revised version.

P3L82 You can remove the ", the Netherlands," after "Eindhoven

It was removed.

P3L84 Can you elaborate quickly why the water is a problem. Is it because of the double periodic boundary, so you assume opposite sides to be of similar nature? Or more specifically a problem to your case study with DEPAC?

The former indeed. We added the explanation (P3L84) in the revised version:

"… since there is no coastline or large water bodies in the domain that could induce secondary circulations."

P4L93 Missing reference

We changed the sentence in the revised manuscript, because the same section was referred to twice.

P4L93:

"Next, we demonstrate the application of the system in a case study for Eindhoven (Section 3)."

We also added a reference to the new section here:

"A first evaluation against $NH_3$ concentration and flux observations at Cabauw is included in Section 4."

P5L145 Missing reference

We added it in the revised manuscript.

P5L149/150 I think you should at least introduce the units of V, G and R once. Probably here. Units are emitted quite often throughout the manuscript. Please add them where useful (e.g. also when you introduce χ)

We reviewed the whole paper, properly introduced variables and added the units of these variables.

P6Fig1 Rext is called Rw throughout the manuscript. In the schematic it shows that Rinc depends on LAI, whereas in Eq. 12 it depends on SAI.

Both inconsistencies have been addressed in the figure.

P6L163 Can you explain why Rb is small when the quasi-laminar layer is small. It is not immediately clear from equation (3).

The quasi-laminar layer is a model for the transition between the roughness on the surface (buildings, grassland, trees) and the free atmosphere. It is assumed laminar and it forms an obstacle to mass transfer from and to the surface. The thickness of the layer scales with ν/u*, where ν is kinematic viscosity. ν can be considered constant (only weak function of T and P) and u* scales with wind speed. The stronger the wind, the stronger the turbulence, and the smaller the layer. Rb represents the resistance of this layer against mass transfer. The stronger the wind, the lower this resistance (Rb ~1/u*).

P7L1 Can you mention how these quantities are actually being calculated within DALES? Are you using the literature values of air etc or the sub-grid quantities from the LES?

As mentioned in the previous comment, it concerns a laminar layer, so the relation needs the molecular diffusivities of heat, momentum and mass.

P7L179 "Absorbed" on the surface is a somewhat strange expression. Maybe use "canopy" instead of "surface" in this paragraph? Or do you mean "adsorbed on the surface"?

We changed the text to use "absorbed through the surface" for clarity, since this is the case for vegetation, soil and water droplets on the underground.

P7Eq5 Are the signs correct, the deposition flux is negative when deposition occurs? ($x_{atm} > x_{comp}$; $V_d$ is always ≥0)

We have reviewed the signs in Equation 5 and confirmed that the deposition flux is negative when deposition occurs, consistent with the convention that $V_d$ is defined as a vector in upward direction.

P8L2 Can you give more information about this "related project"?

This statement was found to be erroneous on further investigation of the related project. The deposition velocity for NO and $NO_2$ only differ by approximately 12% between a completely wet surface vs a dry surface (save for some waterways). The difference for $NH_3$ is much larger. Note that this is a worst case scenario, where the complete surface is saturated with water. In that case, $NH_3$ deposition is controlled by the high solubility of $NH_3$ in water.

We did want to be sure that working only with the equations for dry would be justified. KNMI weather data at the location of Eindhoven Airport showed that the date of our simulations was preceded by a prolonged period of dry conditions (11 days without any precipitation). We have added a line in the discussion of the Eindhoven case study stating this and the statement on the related project was deleted.

In P8L196, we replaced:

"Here, we circumvented this problem by selecting dry days in a period with little or no rain. In addition, a sensitivity analysis in a related project pointed out that the effect of switching between dry and wet land on the deposition fluxes of $NH_3$ is not very strong."

By:

"In our case study, we circumvented this problem, since the KNMI weather data at the location of Eindhoven Airport showed that the date of our simulations was preceded by a prolonged period of dry conditions (11 days without any precipitation)."

P8L207 line starts with ","

Removed the comma.

P8L208 can you elaborate why phenology is negligible

Van Zanten et al (2010) have shown this in their description of the DEPAC model. They show that, during the growing season, the phenology correction factor is either 1 or very close to 1. This only holds for the land use classes currently covered in DEPAC. Moreover, the major land use types in our study domain are urban, arable land and grassland. Only for arable land, the phenology correction factor deviates from 1, but outside the time range of our study (i.e. outside the growing season). We will clarify this in our paper.

The seasonal dependence of $R_{stom}$ is covered by a dependence on LAI, however. This was not clear from Eq. 6, so we will add it as a separate equation.

We will change P8L208:

"where the correction factor for phenology $f_{phen}$ and the correction factor for soil water potential $f_{swp}$ are both assumed equal to 1.0, since the influence of phenology is negligible for the land use classes in DEPAC and soil water potential is expected to be of limited influence in our study area. The seasonal dependence is covered by the use of a leaf area index that varies with the growing season of the vegetation."

To:

"where the correction factor for phenology $f_{phen}$ and the correction factor for soil water potential $f_{swp}$ are both assumed equal to 1.0. This assumption is justified as follows: for the current LU classes, the impact of phenology is minimal (Van Zanten et al., 2010). Instead, the seasonal variation in vegetation phenology is accounted for by using a leaf area index (LAI) that changes according to the vegetation's growing season.

$$G_s^{max} = g_s^{max} \cdot LAI$$

where $g_s^{max}$ is the maximum leaf conductance. Similarly, the influence of soil water potential in North-Western Europe is expected to be limited (Van Zanten et al., 2010)."

P8L208 In this study it might not be relevant, but what about another study? Are the correction factors implemented but not used here, or are they not implemented?

They are not currently implemented, and not relevant to our case study. We will implement these factors when we will apply the model to areas outside of NW-Europe.

P8L209 Please point at section 2.3.2 where you describe the seasonal dependence, i.e. how LAI affects Rinc,

Note that Rstom is a function of LAI, but in a different way from Rinc. This was not clear from the text, so we will add the Rstom dependence on LAI as a new equation (see also the comment above).

P8Eq8 Suddenly you indicate that Gamma is a function of the surface temperature, but in other equations you don't (e.g. Eq 7 or Eq 10). Please be consistent.

We have ensured consistency in indicating that Gamma is a function of surface temperature across all relevant equations, including Equations 7 and 10.

P8L216 what does "long term" mean?

Basically, this is the background concentration at 4 m. Usually, a monthly mean value is taken for '$\chi_{a,4m, long term}$'. The reason for using this mean value is now clarified in the text by adding these lines (at P8L217):

"This approach was chosen, because it is unknown how much of the depositing species is accumulating in the vegetation; there is no mass balance of the vegetation. Instead, the compensation point is estimated by assuming the accumulated amount of a species in the vegetation is in equilibrium with its long term mean atmospheric concentration."

P8L219 missing space between 2 and "s m$^{-1}$"

We will add a space here.

P9L225 Which value is assumed for Rw

A value of 200 s m$^{-1}$ is assumed for frozen soil. We will add it here.

P9L226 I would rephrase and reference Eq. 7 "..calculated for external surfaces analogue to Eq 7 by substituting GammaW for GammaS"

We will rephrase as suggested:

"A compensation point can be calculated for the external surfaces analogue to Eq. 7, by substituting Γw for Γs"

P9L228 is the compensation point at 4m not "long term" anymore, what is it then?

See the comment above about the stomatal compensation point. In the case of external surfaces, there is no volume to accumulate ammonia, so a long term average is not needed. Instead, it is assumed there is always a thermodynamic equilibrium between the surface and its surroundings.

Added P9L228:

"For external surfaces, it is assumed that there is always a thermodynamic equilibrium between the surface and its surroundings. Therefore, $\chi_{a,4m}$ has the same value as the atmospheric concentration."

P9Eq12 case 2: Can u* be negative?

In fact, only the equal sign is important here. The smaller than sign is present in the code to prevent occasional rounding errors in the calculation of u* from crashing the application. We will apply the equal sign in the text.

P9L239 the formula is unclear, what are aSAI and bSAI, why is there a bracket around +bSAI? Please clarify.

There is an error in the equation, which we have corrected. It should say SAI = aSAI * LAI + bSAI. This was a parametrization added to the implementation of DEPAC to enable calculation of SAI from LAI.

P9L245 "Equation 7" should be "Eq. 7"

This was adapted as suggested.

P9L246 I cannot find the compensation point or Gamma in Table A1

We have added it to the table with DEPAC parameters. We have included the LSM and DEPAC parameters in separate tables (A1 and A2, respectively).

P10L255 Are these all from ERA5?

Yes they are. It is mentioned in line 259.

P10L255 what does "partly" mean, how are the other forcings implemented?

Some variables are implemented as actual values (e.g. temperature, humidity, wind) and other as tendencies of these state variables. We have clarified this in the revised version.

P10L256 The "surface energy balance calculation"

We have added "surface" to this sentence.

P10L265 It is not clear at this point that the 1 km x 1 km is the resolution of the emission inventory and 50 m x 50 m refers to the grid size in DALES

We have clarified this in the revised version:

"This fraction map describes the fraction of the emissions in each $1 \times 1$ km$^2$ grid cell of the inventory that is assigned to the $50 \times 50$ m$^2$ grid cells of DALES."

P10L266 move the sentence about the 12 fraction maps to line 279, after the fraction maps are actually described. "for several other source type (CBS, 2023). This results in a total of 12 fraction maps"

We have moved the sentence as suggested.

P10L280 "emission heights"

Adapted as suggested.

P10L281 how high is "ground level"?

In the DALES simulations in this work, a layer height of 20 m was applied near the ground. Emissions at ground level are thus emitted into this layer. This is clarified in the text of the revised version.

P11L284 I don't understand the argument about why road emissions are smaller than highway emissions? Wouldn't we expect them to be smaller, what's the ratio in the initial emission inventory? What is the problem with the downscaling? Does it affect other emission categories? Can you please elaborate on this point

The emissions we used here as basis for the downscaled emissions are from the Dutch Emissie Registratie (Emission Registration) at $1 \times 1$ km$^2$. It gives emissions for several road types (highways, main roads, residential roads) separately. Our emission downscaling tool only redistributes the emissions for each road type over each cell of the DALES grid by total read length for each type. This means that the downscaled emissions for the different road types reflect the ratio in the initial inventory. Downscaling the emissions does not affect these ratios for road transport or other emission categories.

The rationale for the much higher $NO_x$ emissions on highways is as follows: a large share of vehicle kilometers is driven on highways, even though their total length is much shorter than that of secondary roads. This leads to relatively high emissions being attributed to highways. Diesel vehicles, especially trucks, have much higher $NO_x$ emission factors than gasoline cars. Since trucks are more common on highways, this further increases $NO_x$ emissions on highways compared to secondary roads.

P11L293 the large-scale wind direction was easterly? Because Figure 4 shows the local winds to be north/north-easterly

Northeast may be more appropriate here. The ERA5 wind fields that represent the large-scale wind direction are from this direction (between 30 and 70°). Of course, local deviations from the large scale wind direction are possible due to, for instance, land surface effects. The observations near the surface indeed show north/north-easterly winds.

We will change 'east' to 'northeast' in this sentence.

P11L297 The formula or numbers are wrong: 20 m*1.009^127 is only 62 m and not 95 m. Also, the cumulative sum is not 8.5 km.

This is a typo: the number of layers should be 176. This was corrected in the revised version.

P12L320 It seems ERA5 has the dimensions swapped? Should it be 28 km x 17 km?

That is correct. We have adapted this in the revised manuscript.

P12L324 Remove the "since", the STD is not solely a consequence of the rounding.

Changed the text to "at least 0.29 ms-1 and 3deg". The rounding error is expected to be one of the largest errors in the measurement of wind speed and direction, so the estimate is probably not far off the true value.

P12L325 replace "quantized" with "rounded". "The wind speed is rounded down to the next integer value and the wind direction is rounded down to the next full degree."

We've used 'rounded to the nearest integer value / nearest full degree' instead of 'quantized' in the revised version.

P12L319 at which height are these quantities in DALES and ERA5?

In Dales these quantities are at the lowest model level, which is from 0 to 20 m above ground level. For ERA5, we used the values as interpolated to the DALES grid, so they are also representative of 0-20 m.

P12L325 "standard deviation of the measurement errors". I don't understand what exactly you mean by this. The measurement error? The error introduced by rounding? The standard deviation between the model and the measurement?

We now use the term 'precision' here, since it better reflects the quantity. Indeed, here we only mention the error by rounding.

P12L327 °C or K instead of "degree"

We have replaced 'degree' by 'K'.

P12L327 It's the other way round, the model has a negative bias during the day

Thanks for pointing out this error, we adapted it.

P12L327 Does DALES not offer a parameterised 2m-temperature? DALES follows quite closely the ERA5, how do they compare in terms of height/layer thickness?

DALES does not offer a parameterized 2m temperature by default. However, we have included a new figure in which the DALES outputs are 1) extrapolated to measurement height (1.5 m) using MOST and 2) sampled at the measurement location.

ERA5 data is here interpolated to the DALES vertical grid using the LS2D package, and shown for the lowest layer (i.e. 10m).

The figure shows that in general, DALES follows the trend in the ERA5 forcings, also when DALEs is sampled at the measurement location. However, the virtual potential temperature is strongly underestimated to both ERA5 and the observations before and after sunrise. This points at possible surface balance issues at the specific location, in addition to the general difficulty in representing stable boundary layer conditions. The details of this are beyond the scope of this work, so we highlight these issues for future studies.

We adapt the text at P12L326 from:

"There is a slight difference with the observations. The virtual potential temperature shows a 1-2 degree deviation, with a positive model bias during daytime and a negative bias after sunset. This is likely caused by the difference between the lowest model level (from 0 to 20 m above ground level) and the measurement height (2m). The specific humidity matches within 0.002 kg kg$^{-1}$, but there is a 2-3 hr shift of a

trough in the morning. Wind speed shows stronger deviations: up to 2 m s−1 speed differences."

To:

"The $\theta_v$ is represented well by DALES between 06:00 and 18:00 LT, but strongly underestimated before and after sunset. The $q_t$ matches within 0.0015 kg kg−1, but there is a 2-3 hr shift of a trough in the morning. Wind speed shows stronger deviations: up to 2 m s$^{-1}$ speed differences. The wind direction in DALES and ERA5 show an northeast to east direction during most of the day while the observations are between north and northeast."

[Figure]

*Figure 3 (Fig. 4 in the revised MS): Time series of virtual potential temperature, specific humidity, wind speed and wind direction from DALES (blue), ERA5 (green) and observations (dots) on 16 June 2022. DALES data is extrapolated to measurement height (1.5 m).*

P13Fig4 When comparing model to measurement, you should probably extract the value at the measurement location or interpolate and not use domain averages.

We have included model results at the measurement location in the revised version, see also the previous point.

P13Fig4 WDIR How can there be a negative value, is the axis not from 0° to 360°?

This negative wind direction is in the measurement data, which were obtained from KNMI. Their description of the data reads "Mean wind direction (in degrees) for the

10-minute period preceding the observation time stamp (360=north, 90=east, 180=south, 270=west, 0=calm, 990=variable)". This implies a scale from 0 to 360°. It is unclear how a negative value can occur. However, since it is only 1 datapoint, we decided to simply remove it.

P13Fig4 WDIR: It might help to change the y-axis to go from -180 to +180 degrees, then the artificial "peaks" would disappear. Since the range of wind directions is actually quite limited -90 to +90 would probably also suffice.

Good suggestion, we have adapted the scale of the y-axis.

P14L345 Did you do the simulations at 25 m resolution? Is the set-up identical otherwise? I think a comparison to the 50 m run would be interesting (in terms of deposition). You could have tried a resolution of 10 m without using the emission downscaling to determine if that indeed is appropriate to simulate the evening transition. Why is the sub-grid model not able to compensate?

We indeed performed the 25 m resolution simulation, which was identical to the 50 m resolution simulation in all other aspects. We included a comparison with that simulation in the revised version.

A simulation at 10m resolution would result in a horizontal grid of 2200 x 1600 grid cells, which is not feasible on our computing cluster.

P14L347 $\Delta\theta/\Delta z$ ?

That is correct, it was adapted.

P14L349 You say the BLH of 0 m is due to the choice of criterion, but most other metrics also show 0 m. So is the BLH at night just wrong?

Possibly, but the LLJ method shows a nocturnal BL depth of 200-250 m. Also, Fig 5a shows a typical stable boundary layer potential temperature profile at 00:00. Therefore, we do not think that the representation of turbulence under stable conditions in DALES is wrong, but rather the diagnostic of the BLH under those conditions. However, we think that it goes beyond the scope of this paper to go much deeper into the methods to determine the stable boundary layer height.

P14L357 Reference to the figure missing.

Figure 6 is referred to in the previous sentence.

P14L357 Emissions at 06:00 LT

We changed 'of' to 'at'.

P14L365 Where are the time-profiles from? Reference

The time profiles are those used by default in the LOTOS-EUROS model (Manders et al., 2017). We added a reference.

P14L368 "The resulting vertical profile of mean NOx concentration over the domain (Figure 7a) shows a trend similar to the boundary layer development (Figure 5d / Figure C1?)."

Good suggestion, we have adapted this sentence accordingly.

P14L369 The NOx peak near the surface is at midnight according to Fig 7a. The concentration at 06:00 are the lowest ones, even though one would expect the maximum at the morning rush hour? This does not match your description. How do you explain the huge concentrations at mid-night, is this realistic?

The morning rush hour peak in the emissions is between 7:00 and 9:00 LT (Fig. 6b), so the minimum in NOx concentration at 6:00 LT is not inconsistent with that.

The large concentrations at midnight are explained by the ongoing emissions into a stable boundary layer. The emissions peak at 18:00 LT and then steadily drop to their minimum at 04:00 LT. The surface concentrations of $NO_x$ and $NH_3$ peak at 22:00 LT, which coincides with a minimum in the diagnosed boundary layer height. At 00:00 LT, the concentrations have slowly decreased from their peak values, due to weak vertical mixing and advection (wind speed of about 2 m s$^{-1}$).

The question is indeed how realistic this is. In previous comments, we have already pointed the issues that DALES currently has in simulating the transition from a convective to a stable boundary layer. We expect that improvements in the representation of that transition will also lead to an improved representation of tracer concentrations. However, within the scope of this MS, which deals with the implementation of a deposition module in DALES, we cannot address this issue in more detail. Instead, we opted for showing a full diurnal cycle including the nighttime, including its challenges, and pointed to several possible pathways for improvement in future studies.

The description in P14L369 does indeed not capture this well. We changed it to:

"The concentration shows a peak near the surface in the morning (09:00 LT), caused by the rush hour emissions (Figure 6b) and the low boundary layer into which these emissions are mixed. A second and much higher peak in the NOx concentration is simulated in the evening (22:00 LT). This peak coincides with a minimum in the diagnosed boundary layer height, and is caused by a lack of mixing simulated by DALES in the evening. Improving this will be a priority for future applications of DALES in air quality and deposition studies."

P15Fig5 Vertical profile of domain averaged …

We have added 'domain averaged' to the caption in the revised version.

P15Fig5 Please adopt the x-axis of (a) and (c) to better reflect the range of the data

We have adapted the scales of the axes.

P16Fig6 Are the time profiles domain averaged? You use "CEST" here but "LT" everywhere else.

The time profiles are indeed domain averages. We have added that in the caption. We also changed 'CEST' into 'LT' for consistency.

P16L381 domain averaged concentrations at 10m

We have added 'in the lowest model layer (between 0 and 20 m)'.

P16L382 What you mean is that during the entire day the emissions and the upward mixing are in a balance until sunset, leading to a relatively constant near-ground concentration? Only slightly elevated above the background of 1ppb. Is this realistic? Again, how do you explain the huge concentrations at night?

Indeed we mean that the near-surface concentration is relatively constant during much of the day. We will reformulate this sentence to express that more clearly.

The question on how realistic the nocturnal concentration peak is, has been addressed in a reply to previous comment.

P19L1 Is the NOx-background the same on all vertical levels?

We prescribed background conditions for NOx and NH3 as follows:

$z < 200m$: [NOx] = 1 ppb, [NH3]= 2.7 ppb

$200<z<=1500m$: [NOx] = 0.5 ppb, [NH3]= 0 ppb

$z>1500m$: [NOx]=[NH3]=0 ppb

This information is mentioned in Section 3 (P12L303-306), so we refer to it in the revised MS.

References: "doi" is sufficient and preferred over "url". "Publisher" is not necessary, "_eprint", some websites lack the date when they were accessed.

We cleaned up the reference list for the revised MS.

**References**

Beare, R. J., Macvean, M. K., Holtslag, A. A. M., Cuxart, J., Esau, I., Golaz, J.-C., Jimenez, M. A., Khairoutdinov, M., Kosovic, B., Lewellen, D., Lund, T. S., Lundquist, J. K., Mccabe, A., Moene, A. F., Noh, Y., Raasch, S., and Sullivan, P.: An Intercomparison of Large-Eddy Simulations of the Stable Boundary Layer, Boundary-Layer Meteorol, 118, 247–272, https://doi.org/10.1007/s10546-004-2820-6, 2006.

Dai, Y., Basu, S., Maronga, B., and de Roode, S. R.: Addressing the Grid-Size Sensitivity Issue in Large-Eddy Simulations of Stable Boundary Layers, Boundary-Layer Meteorol, 178, 63–89, https://doi.org/10.1007/s10546-020-00558-1, 2021.

Deardorff, J. W.: Stratocumulus-capped mixed layers derived from a three-dimensional model, Boundary-Layer Meteorol, 18, 495–527, https://doi.org/10.1007/BF00119502, 1980.

Jongenelen, T., van Zanten, M., Dammers, E., Wichink Kruit, R., Hensen, A., Geers, L., and Erisman, J. W.: Validation and uncertainty quantification of three state-of-the-art ammonia surface exchange schemes using $NH_3$ flux measurements in a dune ecosystem, Atmospheric Chemistry and Physics, 25, 4943–4963, https://doi.org/10.5194/acp-25-4943-2025, 2025.

Manders, A. M. M., Builtjes, P. J. H., Curier, L., Denier van der Gon, H. A. C., Hendriks, C., Jonkers, S., Kranenburg, R., Kuenen, J. J. P., Segers, A. J., Timmermans, R. M. A., Visschedijk, A. J. H., Wichink Kruit, R. J., van Pul, W. A. J., Sauter, F. J., van der Swaluw, E., Swart, D. P. J., Douros, J., Eskes, H., van Meijgaard, E., van Ulft, B., van Velthoven, P., Banzhaf, S., Mues, A. C., Stern, R., Fu, G., Lu, S., Heemink, A., van Velzen, N., and Schaap, M.: Curriculum vitae of the LOTOS–EUROS (v2.0) chemistry transport model, Geosci. Model Dev., 10, 4145–4173, https://doi.org/10.5194/gmd-10-4145-2017, 2017.

Swart, D., Zhang, J., van der Graaf, S., Rutledge-Jonker, S., Hensen, A., Berkhout, S., Wintjen, P., van der Hoff, R., Haaima, M., Frumau, A., van den Bulk, P., Schulte, R., van Zanten, M., and van Goethem, T.: Field comparison of two novel open-path instruments that measure dry deposition and emission of ammonia using flux-gradient and eddy covariance methods, Atmospheric Measurement Techniques, 16, 529–546, https://doi.org/10.5194/amt-16-529-2023, 2023.

Van Zanten, M. C., Wichink Kruit, R. J., van Jaarsveld, H. A., and van Pul, W. A. J.: Description of the DEPAC module : Dry deposition modelling with DEPAC_GCN2010, Rijksinstituut voor Volksgezondheid en Milieu RIVM, 2010.

Wintjen, P., Schrader, F., Schaap, M., Beudert, B., Kranenburg, R., and Brümmer, C.: Forest–atmosphere exchange of reactive nitrogen in a remote region – Part II: Modeling annual budgets, Biogeosciences, 19, 5287–5311, https://doi.org/10.5194/bg-19-5287-2022, 2022.